# Photodamage repair pathways contribute to the accurate maintenance of the DNA methylome landscape upon UV exposure

**Stéfanie Graindorge**, **Valérie Cognat**, **Philippe Johann to Berens**, **Jérôme Mutterer**, **Jean Molinier***

Institut de biologie moléculaire des plantes, UPR2357-CNRS, Strasbourg, France

* jean.molinier@ibmp-cnrs.unistra.fr

**Data Availability Statement:** WGBS, DNA-seq and small RNA-seq raw data generated in this work are

## Abstract

Plants are exposed to the damaging effect of sunlight that induces DNA photolesions. In order to maintain genome integrity, specific DNA repair pathways are mobilized. Upon removal of UV-induced DNA lesions, the accurate re-establishment of epigenome landscape is expected to be a prominent step of these DNA repair pathways. However, it remains poorly documented whether DNA methylation is accurately maintained at photodamaged sites and how photodamage repair pathways contribute to the maintenance of genome/methylome integrities. Using genome wide approaches, we report that UV-C irradiation leads to CHH DNA methylation changes. We identified that the specific DNA repair pathways involved in the repair of UV-induced DNA lesions, Direct Repair (DR), Global Genome Repair (GGR) and small RNA-mediated GGR prevent the excessive alterations of DNA methylation landscape. Moreover, we identified that UV-C irradiation induced chromocenter reorganization and that photodamage repair factors control this dynamics. The methylome changes rely on misregulation of maintenance, *de novo* and active DNA demethylation pathways highlighting that molecular processes related to genome and methylome integrities are closely interconnected. Importantly, we identified that photolesions are sources of DNA methylation changes in repressive chromatin. This study unveils that DNA repair factors, together with small RNA, act to accurately maintain both genome and methylome integrities at photodamaged silent genomic regions, strengthening the idea that plants have evolved sophisticated interplays between DNA methylation dynamics and DNA repair.

## Author summary

Living organisms have to efficiently respond to environmental cues that interfere with different cellular processes. Upon exposure to biotic/abiotic stresses, the coordinated maintenance of genome and epigenome integrity is crucial to allow the accurate progress of the developmental programs. In plants the sunlight used for photosynthesis also induces the formation of photodamage altering DNA structure. Although photolesions repair

publicly accessible through the GEO registration number: GSE132750.

**Funding:** JM was supported by the LABEX NetRNA (ANR-10-LABX-0036_NETRNA) The funders had no role in study design, data collection and analysis, decision to publish, or preparation of the manuscript

pathways are well characterized, the side effect of UV irradiation on epigenome integrity is yet-to-be fully investigated. Using genome wide approaches and several photodamage repair deficient Arabidopsis plants we determined that UV-C irradiation induces alterations of DNA methylation landscape and that all photodamage repair pathways contribute to the accurate maintenance of methylome integrity predominantly in silent genomic regions. These UV-induced methylation changes are accompanied by the modulation of constitutive heterochromatin organization. Moreover, our study highlighted that photolesions are source of DNA methylation alterations strengthening the idea that complex interplays between DNA damage, DNA repair and DNA methylation dynamics exist.

## Introduction

DNA carries the genetic information that living organisms decrypt to efficiently ensure their developmental programs and their response/adaption to environmental cues. Exposure to biotic/abiotic stresses can directly or indirectly induce the formation of DNA damage such as bases modifications, DNA breaks, alterations of DNA structure all interfering with DNA replication and transcription [1]. Due to their lifestyle, photosynthetic organisms use the beneficial effect of sunlight [2]. However, they have to cope with the damaging effects of specific wavelengths that impact their genome integrity [3]. Indeed, ozone filtered ultraviolet (UV 315–400 nm) and infra-red (IR > 700 nm) induce different types of genomic alterations such as DNA damage, Transposable Elements (TE) mobilization and transposition [4, 5, 6]. In order to maintain genome stability, DNA repair pathways and tight suppression of transposition need to be efficiently activated to prevent DNA sequence alterations and/or genome rearrangements [7, 8, 9]. UV-B (environmental wavelength) and UV-C (experimental wavelength) directly react with DNA bases to produce photolesions [10, 11]. The induced photodamage are cyclobutane pyrimidine dimers (CPDs) and 6,4-photoproducts (6,4 PP; [4]). Both types of photolesions are formed between di-pyrimidines (TT, CC, TC and CT; [3]) and are localized in euchromatin and heterochromatin [12].

In plants, UV-induced DNA lesions are preferentially repaired by Direct Repair (DR) that is a light-dependent error-free process catalyzed by enzymes called DNA photolyases [13]. In *Arabidopsis thaliana*, two active photolyases act specifically on photolesions: PHR1 on CPDs and UVR3 on 6,4 PP [4]. Conversely, a light-independent mechanism, called Nucleotide Excision Repair (NER), repairs photolesions by the excision of the damaged DNA strand, followed by restoration of an intact double helix through *de novo* DNA synthesis [14]. NER is subdivided into two sub-pathways: Transcription-Coupled Repair (TCR) and Global Genome Repair (GGR), that process photolesions along actively transcribed DNA strands or throughout the genome, respectively [14]. The recognition of photolesions during TCR and GGR differ whereas the following steps, DNA unwinding, excision, gap filling and ligation share similar factors [14]. In actively transcribed genomic regions, the stalled RNA POLYMERASE II (RNA POL II) triggers the recognition signal that allows recruiting the COKAYNE SYNDROME proteins A and B (CSA, CSB; [14]. Conversely, during GGR, the DNA DAMAGE-BINDING PROTEIN 2 (DDB2) recognizes the UV-induced DNA lesions in un-transcribed or weakly transcribed genomic regions [14, 15].

Interestingly, DDB2 also associates with the silencing factor ARGONAUTE 1 (AGO1) to form a chromatin-bound complex with 21-nt small RNA (siRNAs, [16]). This class of small RNAs, called UV-induced siRNA (uviRNAs), originates from the photo-damaged regions (mainly TE and intergenic regions) and involves a non-canonical biogenesis pathway

requiring the plant specific RNA POLYMERASE IV (RNA POL IV), RNA-DEPENDENT RNA POLYMERASE 2 (RDR2) and DICER-LIKE 4 (DCL4, 16]). The DDB2-AGO1-uviRNA complex loads on chromatin upon UV irradiation and likely facilitates photo-damage recognition in an RNA/DNA complementary manner [16]. This recently unveiled DNA repair pathway is called small RNA-mediated GGR.

5-methyl cytosine (5-mC) is a base modification that is a component of the epigenome contributing, with histones post-translational modifications (PTM), to the silencing of TE and to the regulation of gene expression [17]. In plants, cytosines are methylated in the symmetric CG, CHG and asymmetric CHH sequence contexts (where H = A, T, or C; [18]). Upon DNA replication, DNA methylation status of the newly synthetized DNA strand needs to be properly maintained [18]. In Arabidopsis, the methyl moiety is deposited on cytosine by 4 DNA methyltransferases: METHYLTRANSFERASE 1 (MET1), CHROMOMETHYLASE 3 (CMT3), CHROMOMETHYLASE 2 (CMT2) and DOMAINS REARRANGED METHYL-TRANSFERASE 2 (DRM2) [18]. These enzymes specifically maintain DNA methylation in the CG (MET1), CHG (CMT3) and CHH (CMT2 and DRM2) sequence contexts [18]. Additionally, cytosines can be methylated *de novo* by the RNA-directed DNA methylation (RdDM) pathway [19]. This process involves two plant-specific RNA POLYMERASES, RNA POL IV and RNA POL V [19]. RNA POL IV in association with RDR2 produces dsRNA precursors that are diced into 24-nt siRNAs by DCL3 [19]. These siRNAs are loaded into AGO4, which together with DRM2 are recruited to chromatin by the RNA POL V to methylate DNA in the 3 sequences contexts [19].

The DNA methylation profile is the result of the complex balance between gain (*de novo*), maintenance and loss/removal [20]. Indeed, DNA methylation can be passively lost upon DNA replication due to inefficient maintenance process [21]. Conversely, DNA methylation could be actively removed by specific 5-mC DNA glycosylases [21]. The Arabidopsis genome codes for 4 DNA demethylases: REPRESSOR OF SILENCING 1 (ROS1), DEMETER (DME), DEMETER LIKE-2 (DML2) and DEMETER LIKE-3 (DML3) which counteracts RdDM to prevent the spreading of DNA methylation [21]. Active DNA demethylation is related to Base Excision Repair (BER), highlighting that 5-mC is considered as a modified base like any other DNA lesions and thus strengthens the notion that DNA repair and DNA methylation dynamics are closely related [21].

Interestingly, several studies have uncovered that DNA repair factors control the shaping of the DNA methylation landscape. Arabidopsis plants defective in expression of the Mismatch Repair factor, mutS HOMOLOG1 (MSH1), exhibited heritable DNA methylation changes [22]. Arabidopsis DDB2 loss of function leads to DNA methylation alterations at many repeat loci [23]. Indeed, DDB2 forms a protein complex with AGO4 and ROS1 that controls *de novo* DNA methylation and expression/activity of ROS1 [23, 24]. Moreover, depletions of cognate GGR factors in plants and in mammals also lead to alterations of DNA methylation profiles at particular loci [25, 26]. Collectively, these studies robustly support the idea that direct interplays between DNA repair and DNA methylation dynamics exist [3]. Additionally, most of the DNA repair pathways, including NER, are DNA synthesis-dependent repair process [9]. Therefore, the re-establishment of proper DNA methylation landscape at damaged/repaired sites is a prominent part of these pathways that should not be under estimated. Ultimately, maintenance of genome and methylome integrities have to be considered as mechanistically interconnected.

The consequences of environmental cues on DNA methylation landscape as well as the putative role of DNA methylation/demethylation-related factors in response to biotic/abiotic stresses have been extensively reported [27, 28, 29, 30]. Biotic and abiotic stresses exposures alter DNA methylome to different extents, leading to the modulation of gene expression, thus

reflecting that DNA methylation mediates response to environmental stress [27, 31, 32]. However, it must also be taken into consideration that most of these stresses also induce DNA damage such as oxidatively-induced DNA modifications, Single and Double Strand Breaks (SSB and DSB) that need to be repaired in order to maintain genome integrity [3, 33]. The effect of genotoxic stress exposure on methylome integrity are yet-to-be fully investigated. Therefore, a major challenge would be to assess whether DNA damage could be sources of DNA methylation changes or not, and also to decipher to which extent particular DNA repair pathways could contribute to control methylome integrity not only genome wide but specifically at damaged sites and upon repair. To uncover this, we took advantage of plants that have to efficiently cope with the deleterious effects of UV radiation and that have evolved sophisticated interconnections between DNA methylation dynamics and DNA repair [3].

In this study, we used genome wide approaches to identify that UV-C irradiation leads to DNA methylation changes predominantly in asymmetric context. These changes are concomitant with the release of silencing of particular repeats and with alterations of chromocenters organization. We unveiled that the DNA repair pathways involved in the repair of UV-induced DNA lesions, namely, DR, GGR and small RNA-mediated GGR, prevent excessive alterations of DNA methylation upon UV-C irradiation. The methylome changes rely on the misregulation of maintenance, *de novo* and active DNA demethylation pathways highlighting that maintenance of genome and methylome integrities are interconnected. The genome wide mapping of UV-C induced photolesions revealed their predominant locations at centromeric/pericentromeric regions. The cross-comparison of methylome changes and of photodamaged regions allowed identifying that UV-C-induced DNA lesions are sources of DNA methylation alterations. Collectively, our data suggest that DNA repair factors, together with small RNA, act to accurately maintain genome and methylome integrities at damaged sites in repressive chromatin, including both constitutive and facultative heterochromatin.

## Results

### UV-C irradiation induces changes of the DNA methylation landscape

In order to characterize the effect of UV-C irradiation on DNA methylation landscape, we determined by whole-genome bisulfite sequencing (WGBS) the DNA methylome of WT Arabidopsis plants prior and 24h upon UV-C exposure. Importantly, the UV-C dose, the growth conditions and the time points used (see methods for details), were set up to favor an efficient induction of photolesions and to prevent significant changes in developmental phenotypes [34, 35].

Indeed, the photodamage repair is expected to be completed 24h upon UV-C exposure in WT and also in DNA repair deficient Arabidopsis plants, allowing the determination of the methylome landscape including repaired genomic regions.

In WT plants, comparison of the DNA methylation levels prior irradiation with those of 24h upon UV-C exposure revealed around 2,000 Differentially Methylated Regions (DMRs) including 55% of hyper-DMRs and 45% of hypo-DMRs, predominantly in the CHH asymmetric context (Figs 1A and S1A). These UV-C-induced DNA methylation changes are predominantly located within the centromeric and pericentromeric regions (Figs 1B, S2A and S3) mainly overlapping with chromatin state 6 (intergenic regions) and with constitutive heterochromatic states 8 and 9 (GC rich within intergenic regions and TE; S1B Fig, [36]). Importantly, the distribution of chromatin-states containing DMRs significantly differs from their overall distribution in the Arabidopsis genome, highlighting a strong bias for methylation changes within constitutive heterochromatin (S1B Fig).

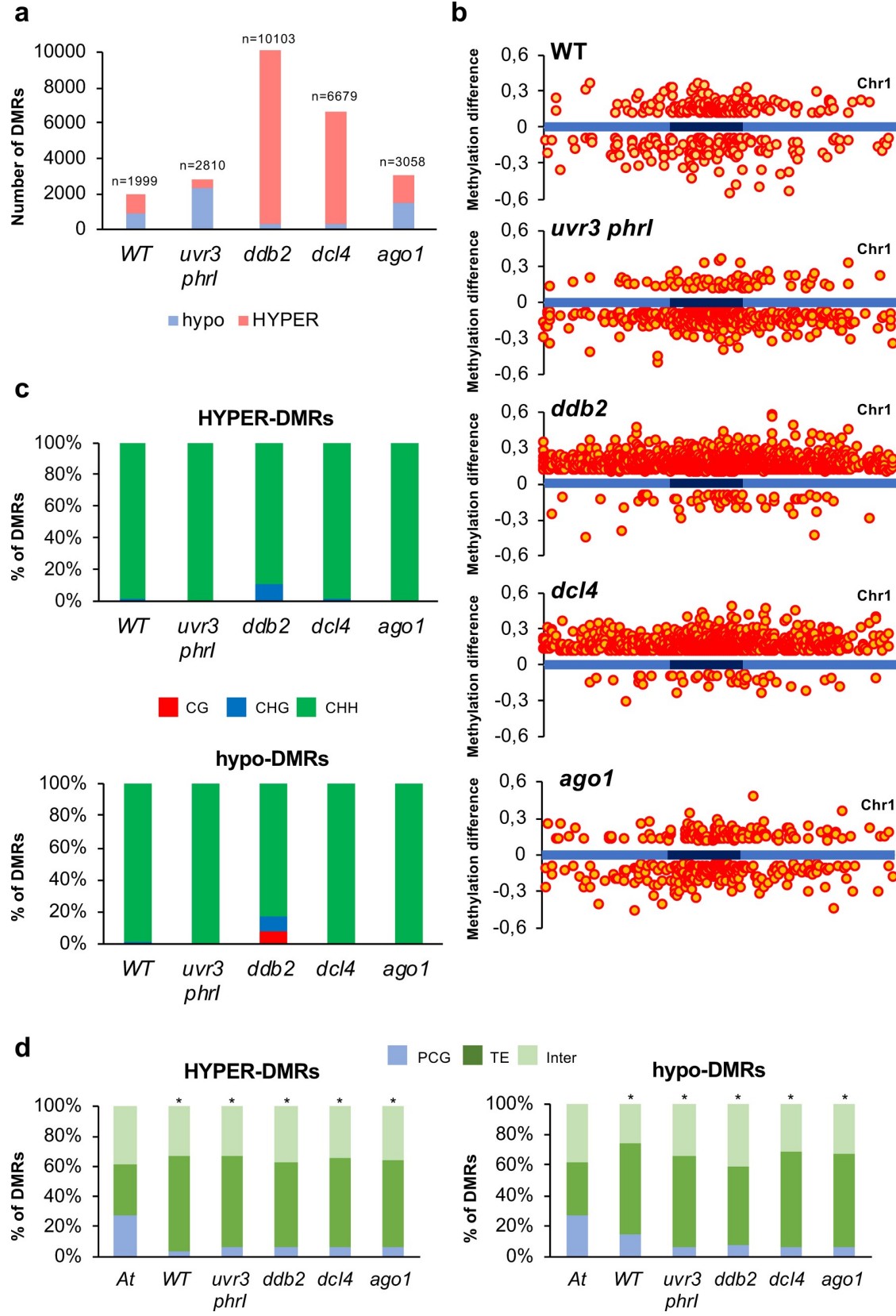

**Fig 1. DNA methylation differences induced by UV-C irradiation. a** Histograms representing the total number of hypo-DMRs (red) and hyper-DMRs (blue) identified in WT, *uvr3 phrI*, *ddb2*, *dcl4* and *ago1* plants 24h upon UV-C exposure. DMRs were calculated relative to their corresponding untreated control. **b** Distributions of DMRs along chromosome 1 (light blue: chromosome arms, dark blue: pericentromeric regions) for the three sequence contexts. Hyper-DMRs and hypo-DMRs are shown above and below each chromosome, respectively. **c** Histograms representing the percentage of CG (red), CHG (blue) and CHH (green) of the hyper- and hypo-DMRs identified in WT, *uvr3 phrI*, *ddb2*, *dcl4* and *ago1* plants 24h upon UV-C exposure. DMRs were calculated relative to their corresponding untreated control. **d** Histograms representing the percentage of the identity (protein-coding genes: PCG, TE and intergenic) of the hyper- and hypo-DMRs identified in WT, *uvr3 phrI*, *ddb2*, *dcl4* and *ago1* plants. *A. t* represents the overall distribution of PCG, TE and intergenic regions in the *Arabidopsis thaliana* genome. * Chi square test < 0.01 compared to *A. t.*

Single resolution analyses of WGBS data allowed identifying that more than 97% of these changes occurred in the CHH context, exhibiting a trend of gain of DNA methylation (Figs 1C and S2B). These DMRs mainly overlapped with TE and intergenic genomic regions (Fig 1D), consistent with their enrichment in silent chromatin states. Interestingly, we could not identify genomic regions with concomitant changes of CHG and CHH methylation, highlighting that alterations of DNA methylation in both contexts are uncoupled in our experimental conditions (Fig 1C). This contrasts with the well characterized positive correlation between CHG and CHH methylation [37, 38] and thus suggests that UV-C irradiation may induce context specific changes of DNA methylation levels.

Collectively, our data showed that, in WT Arabidopsis plants, UV-C irradiation led to balanced gain and loss of DNA methylation in an CHH context that is largely distinct from CHG methylation. Importantly, theses changes in DNA methylation landscape were predominantly located within centromeric-pericentromeric regions representing constitutive heterochromatin.

## Photodamage DNA repair pathways prevent excessive changes of DNA methylation landscape upon UV-C exposure

To assess the putative role of the DNA repair processes in the interplay between maintenance of genome and methylome integrities in response to UV irradiation, Arabidopsis plants deficient for the main pathways involved in the repair of UV-induced DNA photolesions were subjected to UV-C irradiation. DNA methylation profiles were determined by WGBS prior and 24h upon UV-C exposure in order to identify DMRs. We used the double *uvr3 phrI* mutant plants that are defective in both photolyases involved in the direct repair of photoproducts [39]. Therefore, in such plants, the NER pathway (TCR and GGR) would be expected to be the main process used to remove DNA photolesions. We also used *ddb2*, *dcl4* and *ago1* mutant plants that are defective in GGR (*ddb2*, [23, 40]) and/or in small RNA-mediated GGR (*ddb2*, *dcl4* and *ago1*, [16]). Thus, the DR would be expected to be predominantly used to repair photolesions as well as other pathways (i.e. homologous recombination) [33, 41]. Therefore, determining the genome-wide DNA methylation landscapes of these UV-C-treated mutant plants would allow identifying how each DNA repair pathway could, directly or indirectly, contribute to shape the DNA methylome upon UV-C exposure.

Comparative analyses of the DNA methylomes within each genotype, *uvr3 phrI* (0 *vs* 24h), *ddb2* (0 *vs* 24h), *dcl4* (0 *vs* 24h) and *ago1* (0 *vs* 24h) plants, revealed thousands of DMRs (Fig 1A). Indeed, *uvr3 phrI* mutant plants exhibited 2,379 hypo-DMRs upon UV-C exposure, representing more than 84% of the total DMRs (Figs 1A and S1A). Globally, UV-C exposure leads to loss of CHH DNA methylation in *uvr3 phrI* plants (S2B and S3 Figs). Both hyper- and hypo-DMRs are distributed all along the chromosomes arms albeit we can notice an enrichment within the centromeric and pericentromeric regions as observed in WT treated plants (Figs 1B, S2A and S3). Conversely, *ddb2* and *dcl4* mutant plants exhibited 9,750 and 6,350

hyper-DMRs, respectively, representing more than 95% of their total DMRs (Figs 1A and S1A). These observations are consistent with the role of DDB2 in the chaperoning of the RdDM factor AGO4 to control *de novo* DNA methylation [23]. Moreover, it suggests that the small RNA-mediated DNA repair of photolesions, involving DDB2 and DCL4, also controls DNA methylation. Finally, we identified in the *ago1* hypomorphic mutant plants (*ago1-27*, [42]), 3,058 DMRs including 52% of hyper-DMRs and 48% of hypo-DMRs (Figs 1A and S1A). These DMRs are located, like in the other tested plants, mostly in centromeric and pericentromeric regions (Figs 1B and S2A). Globally, UV-C exposure leads to gain of CHH DNA methylation in *ddb2*, *dcl4* and *ago1* plants (S2B and S3 Figs).

Similar to the results obtained in WT plants, the mutant's DMRs mainly overlap with TE (>55% of the total DMRs), intergenic genomic regions (around 30% of the total DMRs; Fig 1D) and with chromatin states (states 4, 5, 6, 8 and 9) that correspond to repressive contexts, as expected (S1B Fig, [36]).

Hyper-DMRs sizes of *ddb2*, *dcl4*, *ago1* plants and hypo-DMRs sizes of *uvr3 phrI* plants are significantly longer than those identified in WT plants (S4A and S4B Fig). This suggests that each DNA repair process restricts the length of regions exhibiting DNA methylation alterations. In addition, in all mutant plants, genomic regions giving rise to hyper-DMRs display higher methylation level prior UV-C treatment compared to WT plants and reciprocally with hypo-DMRs (S4C, S4D and S4E Fig). This would suggest that such regions are more prone to gain or to lose DNA methylation due to the pre-existing influence of particular DNA methylation/demethylation pathways. In all tested mutant plants, DNA methylation changes occurred predominantly in the CHH context although we found, in the *ddb2* mutant, around 10% of the DMRs in the symmetric contexts (CG and CHG; Fig 1C).

All together, these results suggest that, upon UV-C exposure, GGR and small RNA-mediated GGR prevent excessive gain of DNA methylation and that DR prevents excessive loss of DNA methylation. These methylome changes occurred predominantly in repressive chromatin where repeats and TE are abundant.

In WT and *dcl4* plants, LTR/Gypsy TE overlapping with DMRs are significantly over-represented compared to their distribution in the Arabidopsis genome (S1C Fig). In *uvr3 phrI*, *ddb2* and *ago1* plants, class II TE exhibiting DMRs are significantly over-represented compared to those of WT and *dcl4* plants (S1C Fig), suggesting that UV-C irradiation may have triggered TE mobilization. Interestingly, we found that the heat stress responsive LTR/Copia TE, *ONSEN* [43], displayed hyper-DMRs in WT plants upon UV-C irradiation (S5A Fig). This gain of DNA methylation at the edge of the TE as well as in intergenic regions is more pronounced in *ddb2* and *dcl4* plants (S5A, S6A, S6B and S6C Figs) and suggests that UV-C may have released *ONSEN* transcription. We can notice that, in all plants, the 24-nt abundances at the edge of *ONSEN* did not significantly change upon UV-C irradiation (S5A Fig). To test the effect of UV-C irradiation on *ONSEN* transcript level, we measured, 2h and 24h upon UV-C exposure, its RNA steady state level in WT and in DNA repair deficient plants. We found that in WT, *ddb2*, *dcl4* and *ago1* plants *ONSEN* transcripts were up-regulated whereas they were down-regulated in *uvr3 phrI* plants highlighting that UV-C exposure and DNA repair factors modulate its expression (S5B Fig). In agreement with our observations, UV-B stress was reported to precociously release gene silencing of transgene, of TE, as well as of endogenous loci in Arabidopsis and maize [44, 45].

Collectively, our data suggest that UV-C irradiation may have transiently released TEs expression and that changes in DNA methylation could act as a defense mechanism to prevent an additional burst of TE mobilization that would further lead to genome instability. Moreover, it allows considering that DNA repair pathways (DR, GGR and small RNA-mediated GGR) likely contribute to the regulation of the DNA methylation landscape at putative UV-reactivated loci.

## UV response and methylation changes

In order to determine whether UV-C-induced DNA methylation changes are controlled by specific DNA repair processes or whether these methylome alterations result from a general "stress response" effect, we compared the DMRs identified in WT plants with those of each mutant. Additionally, we performed the same cross-comparison using WGBS data from WT plants subjected to drought stress [46]. Interestingly, we found significant overlap between DMRs (hyper and hypo) of WT (UV-C and drought) and of mutant plants as well as in between DNA defective plants as reflected by a representation factor >1 (S7 and S8 Figs). This suggests that DNA methylation levels of particular genomic regions are modulated in a stress-dependent manner and that photodamage repair processes controls the methylome landscape of a common set of regions.

Moreover, we re-analyzed the DNA methylation levels, in the CHH context, of each DMR for all the genotypes. Interestingly, we found that most of the DMRs in *ddb2*, *dcl4* and *ago1* plants display increased DNA methylation levels whilst in *uvr3 phrI* plants these profiles decreased (S9 Fig). These observations are consistent with the respective roles of these factors in the different pathways involved in the repair and in the response to UV-induced DNA lesions [3].

Hence, these data show that methylome changes are part of a general stress-response and that DNA repair pathways act synergistically to maintain DNA methylation landscape integrity in response to UV-C irradiation.

## UV-C irradiation induces chromocenters reshaping

Using WGBS approach, we identified that UV-C irradiation induced genome wide alterations of DNA methylation landscape. Interestingly, many methylation changes overlapped with centromeric and pericentromeric regions suggesting that constitutive heterochromatin strongly reacts to such stress (Figs 1B, S1B and S2A). Given that chromocenters are highly compacted genomic regions, we examined whether UV-C irradiation may have affected their compaction and/or their shape and to which extend photodamage DNA repair pathways act to maintain their shape. For this, we developed an automated image quantification program using DAPI staining and confocal microscopy to determine the percentage of surface occupied by chromocenters in a corresponding nucleus. We defined this parameter as chromocenters occupancy (CO) and determined this value in untreated (time point 0) and UV-C treated leaves (24h upon exposure) of WT and DNA repair deficient plants. In addition, relative chromocenters fluorescence intensity, chromocenters and nucleus surfaces (see methods for details) were determined using the same samples. Interestingly, in WT and *dcl4* plants CO increased whilst it decreased in *uvr3 phrI* and *ddb2* mutant plants and remained unchanged in *ago1* plants (Fig 2A and 2B). UV-C-induced CO alterations are correlated with the significant modulation of chromocenters fluorescence intensity and of chromocenters-nucleus surfaces (S10A, S10B and S10C Fig). This reflects that UV-C irradiation triggers nucleus and heterochromatin reorganization and that DR and GGR factors contribute to different extents to the nuclear dynamics.

In order to determine whether 5-mC distribution follows UV-C-induced chromocenters reorganization and methylome alterations, as characterized by confocal microscopy and WGBS, respectively, we performed immunolocalization of 5-mC. We used untreated (time point 0) and UV-C treated leaves (24h upon exposure) of WT and DNA repair deficient plants. We found in all tested plants, and as expected, that 5-mC localized around chromocenters prior UV-C treatment [47]. We confirmed that 5-mC signal remained unchanged in WT and in *ago1* plants upon UV-C irradiation whereas the signal increased in *ddb2* and *dcl4* plants and decreased in *uvr3 phrI* plants, respectively (Fig 2B). Interestingly, in *ddb2* and in *dcl4* plants,

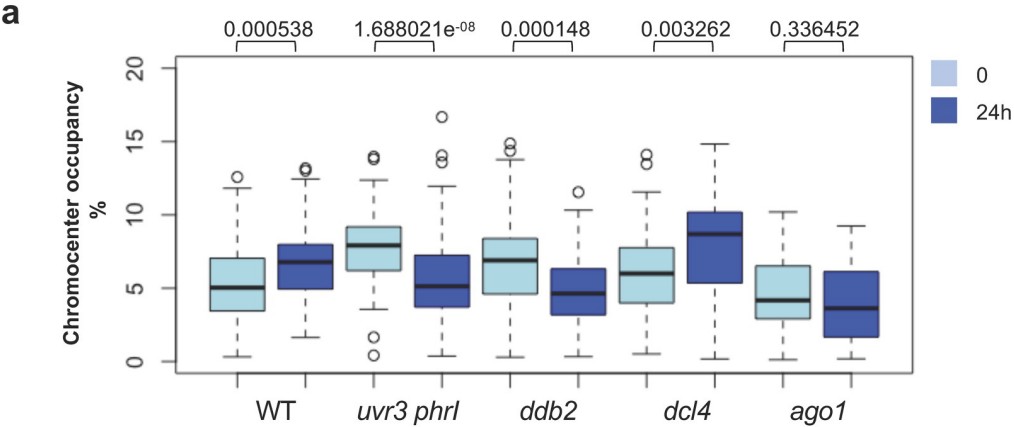

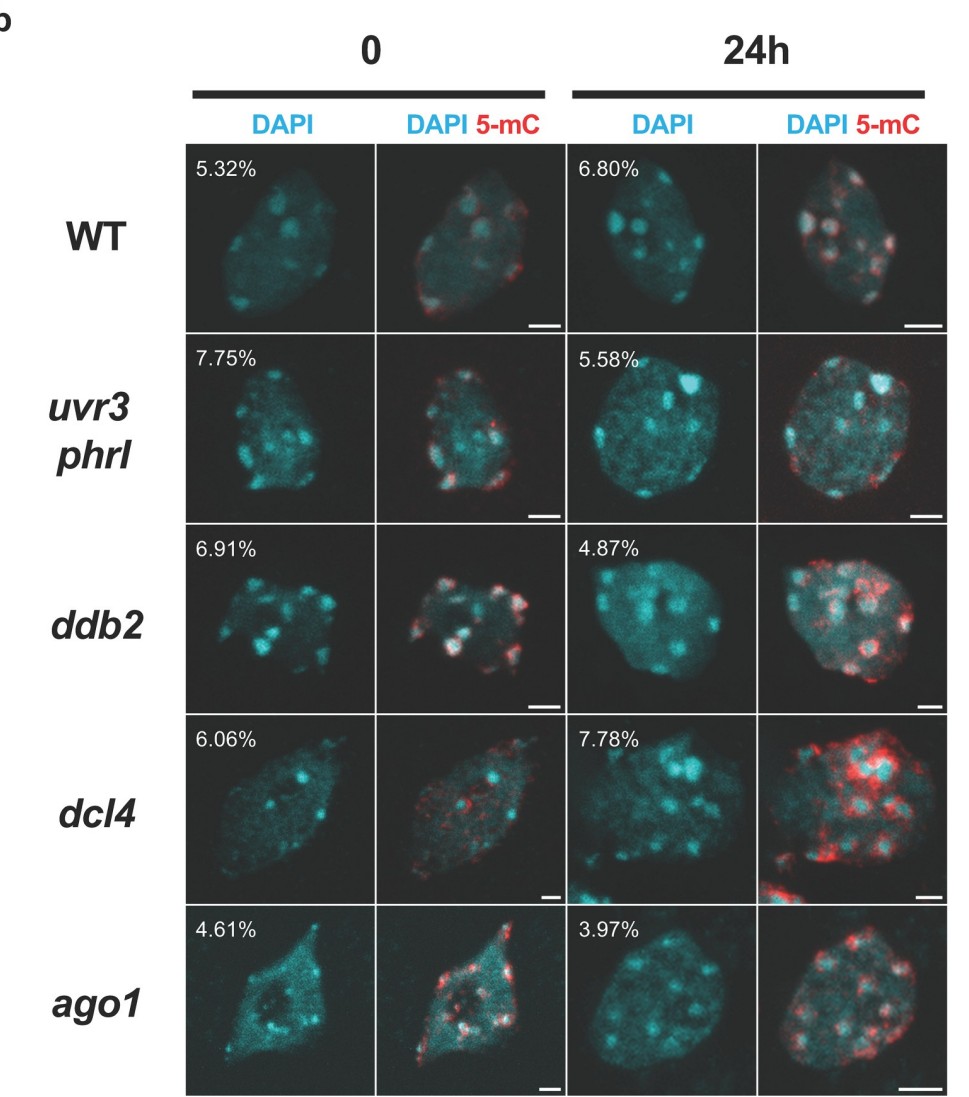

**Fig 2. Chromocenters and 5-mC phenotypes upon UV-C exposure. a** Boxplots representing the percentage of Chromocenters Occupancy (CO) in WT, *uvr3 phrI*, *ddb2*, *dcl4* and *ago1* plants before (0) and 24h upon UV-C exposure. CO was determined using DAPI staining, confocal microscopy and measured with an automated image quantification program. Exact p values according Mann Whitney test are indicated above each graph. Number of nuclei

analyzed: 44 to 109. **b** Chromocenters phenotypes of isolated leaf nuclei of WT, *uvr3 phrI*, *ddb2*, *dcl4* and *ago1* plants before (0) and 24h upon UV-C exposure. Representative DAPI staining (cyan) and 5-mC immunostaining (red) are shown for each genotype and time point. CO are indicated on the representative picture for each genotype and time point. Images were reconstructed from confocal image stacks. Scale bars = 2 μm.

5-mC spread over chromocenters consistent with our WGBS analyses (Figs 1B and 2B). UV-induced modulation of chromocenters shape may lead to transcriptional changes of particular sequences located in this area of the chromosome as already reported for heat-stress [45, 48]. Thus, to test this hypothesis, we analyzed by RT-qPCR the RNA steady state level of the centromeric 180 bp repeats and of the flanking pericentromeric heterochromatic domains containing 5S rRNA before (time point 0) and after UV-C irradiation (2h and 24h). We found that in WT plants the 180 bp and the 5S transcripts levels increased 2h upon UV-C exposure and come back to initial level at 24h (S11A and S11B Fig). In *ddb2*, *dcl4* and *ago1* plants we also measured an increase level of 180 bp and the 5S transcripts levels upon UV-C irradiation whilst in *uvr3 phrI* plants they decreased (S11A and S11B Fig). This shows that UV-C exposure released expression of particular centromeric and pericentromeric sequences and that this response relied on the DR pathway.

Collectively, our data highlighted that UV-C exposure induced the release of silencing of particular repeats as well as structural changes of chromocenters and 5-mC distribution. Moreover, we revealed that DNA repair factors involved in the repair of photodamage contribute to prevent excessive transcriptional reactivation and chromocenters changes in response to UV-C irradiation.

## UV-induced DNA methylome changes rely on misregulation of DNA methylation/demethylation pathways

The methylome analysis of WT and DNA repair defective plants subjected to UV-C irradiation revealed thousands of DMRs (Fig 1A). Such DNA methylation alterations may result from local effects in *cis* and/or from expression changes of genes involved in DNA methylation/demethylation processes. To test the latter hypothesis, we measured by RT-qPCR in WT and DNA repair deficient plants the transcript levels of the main factors of Arabidopsis DNA methylation/demethylation pathways. UV-C induced up-regulation of *CMT2* and *DRM2* in WT, *uvr3 phrI*, *ddb2* and *ago1* plants in agreement with their roles in CHH methylation and with the gain of asymmetric methylation (S12 Fig). Conversely, in *dcl4* plants expression of all DNA methyltransferases was down-regulated whilst enhanced CHH methylation level was measured (S13 Fig). We also found that UV-C irradiation leads to the up-regulation of demethylases expression in all plants except for *DML3* in *uvr3phrI* and for *DML2* in *ddb2* mutant plants (S13 Fig). These results suggest that loss of DNA methylation could result from up-regulation the active DNA demethylation process.

In order to further determine the involvement of DNA methylation/demethylation pathways in these methylome changes, we re-analyzed the DNA methylation levels of each CHH-DMRs using publicly available data of mutant plants defective in the expression of the main Arabidopsis DNA methyltransferases: *met1*, *cmt2*, *cmt3*, *drm1 drm2* [38]. We found that DNA methylation levels at identified DMRs (hyper and hypo) rely mainly on CMT2 and on DRM1/2 DNA methyltransferases, consistent with their roles in maintenance and *de novo* CHH methylation (Fig 3A and 3B, [37]). We found that MET1, involved in maintenance of DNA methylation in the CG context, also influenced CHH methylation at few loci (Fig 3A). Indeed, in *met1* mutant plants many CHH hyper-DMR have been identified [38]. In agreement with this, we found that *MET1* RNA steady state level decreased in WT, *ddb2* and *dcl4* plants

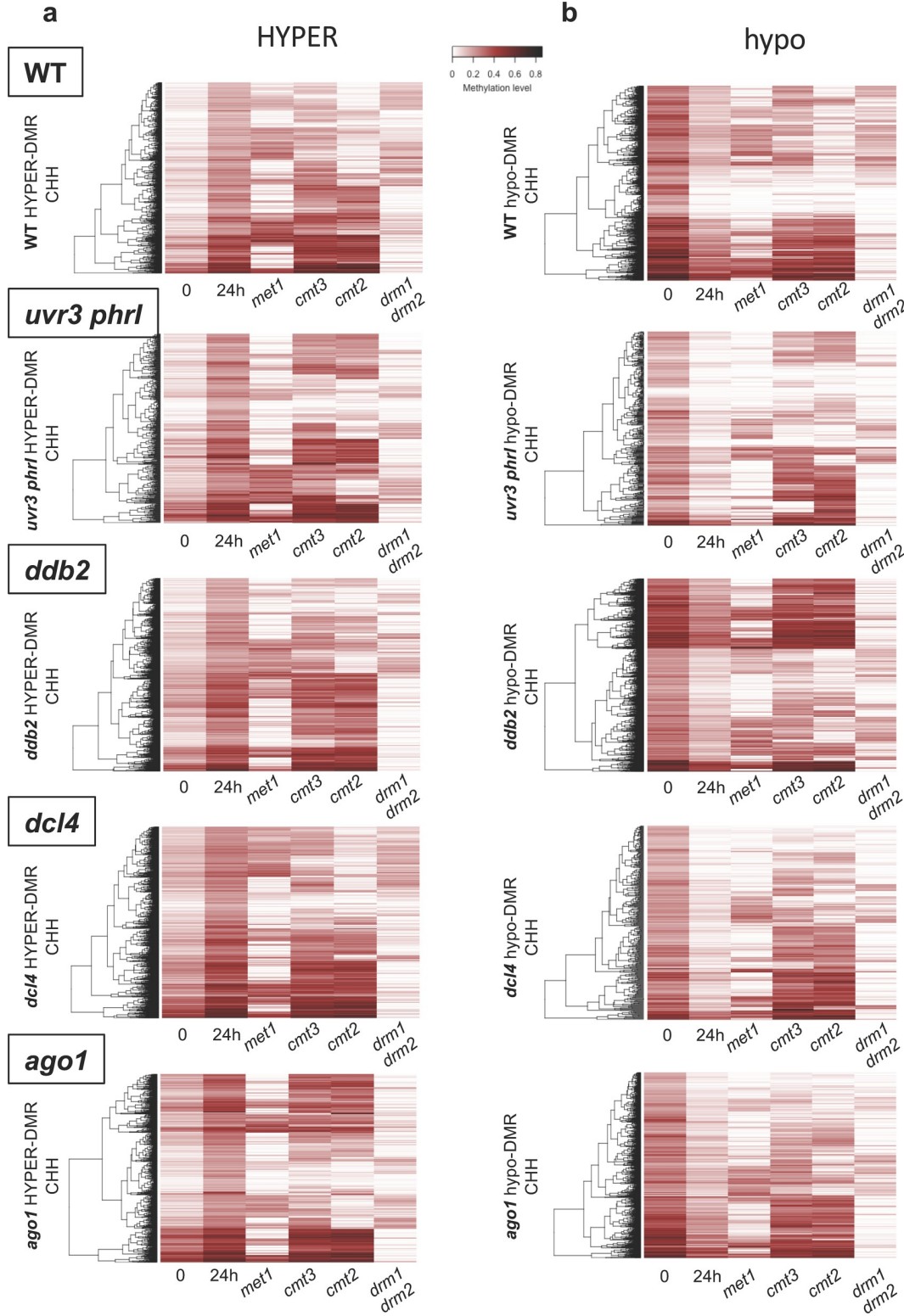

**Fig 3. UV-induced DMRs and DNA methyltransferases.** Heatmaps of CHH methylation levels within hyper-DMRs **(a)** and hypo-DMRs **(b)** identified in WT, *uvr3 phrI*, *ddb2*, *dcl4* and *ago1* plants 24h upon UV-C exposure. The CHH methylation levels of each of these hyper-DMR are reported for *met1*, *cmt2*, *cmt3* and *drm1/2* mutant plants. Columns represent data for each indicated genotype (white: 0; black: 1).

2h upon UV-C exposure supporting the idea that transient down regulation of *MET1* expression may have also contributed to an ectopic gain of CHH methylation (S12 Fig).

Surprisingly, in all tested plants we could not identify genomic regions with concomitant alterations of CHG and CHH methylation levels, strengthening the fact that changes of DNA methylation in both contexts are uncoupled (Fig 1C). These observations are in agreement with the data obtained in *met1* and *ddm1* mutant plants for which alterations of CHH methylation occurred at distinct sites compared to changes of CHG methylation [38].

We observed that DRM1/2 play a predominant role in the regulation of CHH DNA methylation levels at hyper- and hypo-DMRs upon UV-C exposure (Fig 3A and 3B). In order to further determine whether the canonical RdDM pathway is involved in this regulation, we analyzed the DNA methylation levels at hyper-/hypo-DMRs in *nrpd*1 and in *ago4* deficient plants [38]. Indeed, we confirmed that the RdDM pathway plays a major role in the regulation of CHH DNA methylation landscape at most identified DMRs (Figs 4A and S14A). These observations suggest that UV-C exposure induces RdDM dysfunction either stimulating *de novo* or altering maintenance of DNA methylation. Moreover, our results suggest that photodamage DNA repair pathways are, directly or indirectly involved in *de novo* and in DNA methylation maintenance processes.

Active DNA demethylation also leads to loss of DNA methylation [21]. We aimed at determining whether genomic regions exhibiting loss of DNA methylation are targeted by the active DNA demethylation pathway. For this, we compared our identified hypo-DMRs with hyper-DMRs found in plants defective in *ROS1*, *DML2* and *DML3* demethylases expression (*rdd* mutant, [38]). We observed between 35 and 55% overlap with *rdd* hyper-DMRs (S15A Fig) confirming that active DNA demethylation may have acted at these particular loci to reduce DNA methylation level upon UV-C exposure.

Collectively, these analyses revealed that alterations of DNA methylation could have as origin an UV-C-induced misregulation/dysfunction of actors of the DNA methylation pathways (*de novo* and maintenance) and/or of the active DNA demethylation process.

### Small RNA populations upon UV-C exposure

To further decipher the role of the RdDM in the UV-induced regulation of DNA methylation landscape, we determined how canonical small RNA populations (21-, 22- and 24-nt) varied at DMRs. We first determined 24-nt siRNA abundance at hyper-/hypo-DMRs for all plants. In all the cases, and as expected, 24-nt siRNA population abundance is strongly reduced in RNA POL IV deficient plants (Figs 4B and S14B) strengthening the observation that canonical RdDM is involved in the biogenesis of these 24-nt siRNA. Surprisingly, only *ddb2* plants exhibit significant increase of 24-nt siRNA abundance at TE hyper-DMRs (Fig 4B) consistent with the role of DDB2 in the control of *de novo* DNA methylation [23]. Conversely, in all other mutant plants, we could either measure decreased or stabilization of small RNA abundances contrasting with the direct correlation between 24-nt siRNA quantity and gain of DNA methylation (Fig 4B, [19]). For hypo-DMRs, 24-nt siRNA abundance significantly decreased in WT, *uvr3 phrI* and *ago1* plants at TEs whereas no significant changes could be measured in *ddb2* and *dcl4* mutant (S14B Fig). Hence, our data suggest that either 24-nt siRNA abundances have been precociously and transiently modulated/used upon UV-C exposure and/or that other populations of small RNA (21-nt and 22-nt) may play a role in the regulation of DNA methylation. Using publicly available small RNA libraries [16] we found that in WT plants, 30 min following UV-C exposure, 24-nt siRNA abundance increased compared to the untreated time point 0 (S15B Fig) reflecting that transient enhancement of this population of small RNA may have occurred. To test the putative role of 21-nt and 22-nt siRNA in DNA methylation

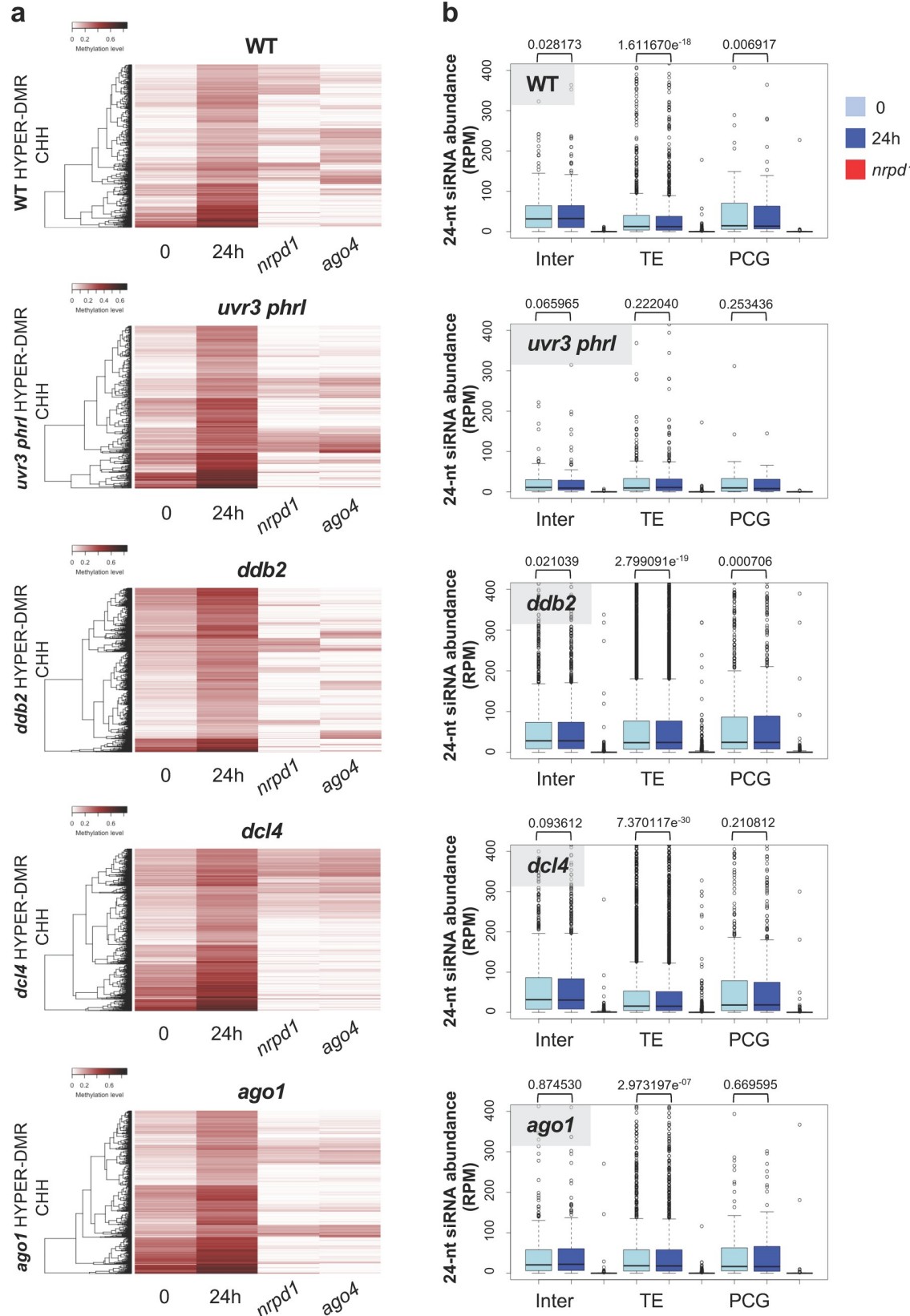

**Fig 4. RNA-directed DNA methylation and DMRs. a** Heatmaps of CHH methylation levels within hyper-DMRs identified in WT, *uvr3 phrI*, *ddb2*, *dcl4* and *ago1* plants before, 24h upon UV-C exposure. The CHH methylation levels of each of these hyper-DMR are

reported for RNA POL IV (*nrpd1*) and AGO4 (*ago4*) deficient plants. Columns represent data for each indicated genotype (white: 0; black: 1). **b** Boxplots representing the abundance of 24-nt siRNAs mapping to the CHH hyper-DMRs identified in WT, *uvr3 phrI*, *ddb2*, *dcl4*, *ago1* and plants. For each genotype the abundance of 24-nt siRNAs is shown in RNA POL IV deficient plants (*nrpd1*). The 24-nt siRNA abundance is normalized against global small RNA content and expressed as reads per million (RPM). p-values are calculated according to Wilcoxon Matched-Pairs Signed-Ranks.

changes we measured their abundances at hyper-/hypo-DMRs. Firstly, we noticed that both 21- and 22-nt siRNA abundances are strongly reduced in RNA POL IV deficient plants (S16A, S16B, S17A and S17B Figs) in agreement with the existence of non-canonical small RNA biogenesis pathways [49]. Secondly, in WT, *uvr3 phrI*, *ddb2* and *dcl4* mutant plants, 21-nt abundance increased for all types of genomic regions exhibiting hyper-/hypo-DMRs whilst in *ago1* plants it decreased (S16A and S17A Figs). The 22-nt siRNA patterns are similar to those of 21-nt siRNA in *uvr3 phrI*, *ddb2* and *dcl4* mutant plants for hyper-/hypo-DMRs whereas their trends differ in WT plants (S16B and S17B Figs) strengthening the idea that DNA methylation may require different populations of small RNA [50]. 21- and 22-nt siRNA originating from up-regulation of TE RNA steady state level and of their degradation products are incorporated into a cognate AGO protein, namely AGO6, to target DNA methylation [51]. We found that DNA methylation level of several loci are under the control of both AGO4 and AGO6 consistent with their redundant functions (S18A and S18B Fig, [51]). Hence, our analyses suggest that POL IV-dependent siRNA (21-, 22- and 24-nt) mediate changes in DNA methylation landscape through complex interplays.

## Genome wide mapping of UV-C-induced photolesions

UV-C exposure leads to the formation of photoproducts at di-pyrimidines [4]. In order to determine a genome-wide map of photolesions, genomic DNA was prepared from the same UV-C treated plants (WT, *uvr3 phrI*, *ddb2*, *dcl4* and *ago1*; see methods for details) used for the methylome analyses. Genomic DNA was subjected to immunoprecipitation using anti-CPD and anti-6,4 PPs antibodies and high throughput sequencing was performed on the immunoprecipitated DNA [16]. The resulting sequences were mapped on the nuclear Arabidopsis genome and the enriched genomic regions (IP/input) containing photolesions were identified using bioinformatic workflow (see methods for details, [16]). We found more than 3,500 regions enriched in photolesions in WT plants whereas these numbers are much more reduced in all tested mutant plants (Fig 5A). This could be due to the physiological adaptation of DNA repair deficient plants which are prone to produce more UV-screen compounds to prevent excessive photo-damage and also to the reduced leaves size of the *ago1-27* plants [39, 42]. In all tested plants, photolesions are located all along the chromosomes with a bias for TE and intergenic regions in mutant plants compared to WT plants (Fig 5B). Importantly, we can observe enhanced IP signals at centromeric/pericentromeric regions, suggesting that this part of the genome is more prone to form photodamage (Figs 5C and S19). Given that photolesions are formed between di-pyrimidines (CC, CT, TC and TT) we calculated their frequencies for each DNA strand of the identified damaged regions (intergenic, TE and Protein Coding Genes: PCG) and compared them to their overall frequencies in the Arabidopsis genome. We could not identify a significant bias for di-pyrimidines frequencies at damaged sites suggesting that other genomic or even epigenomic features may favor photolesions formation (S20 Fig; S1 Table). In order to characterize the epigenomic features of the photo-damaged regions we determined their overlap with chromatin states [36]. In WT plants we found a slight but significant enrichment of the photoproducts in chromatin states 2, 4, 5 containing high repressive H3K27me3 level, representative of facultative heterochromatin, in chromatin state 6 and in repressive chromatin states 8 and 9 representative of constitutive heterochromatin (S21 Fig).

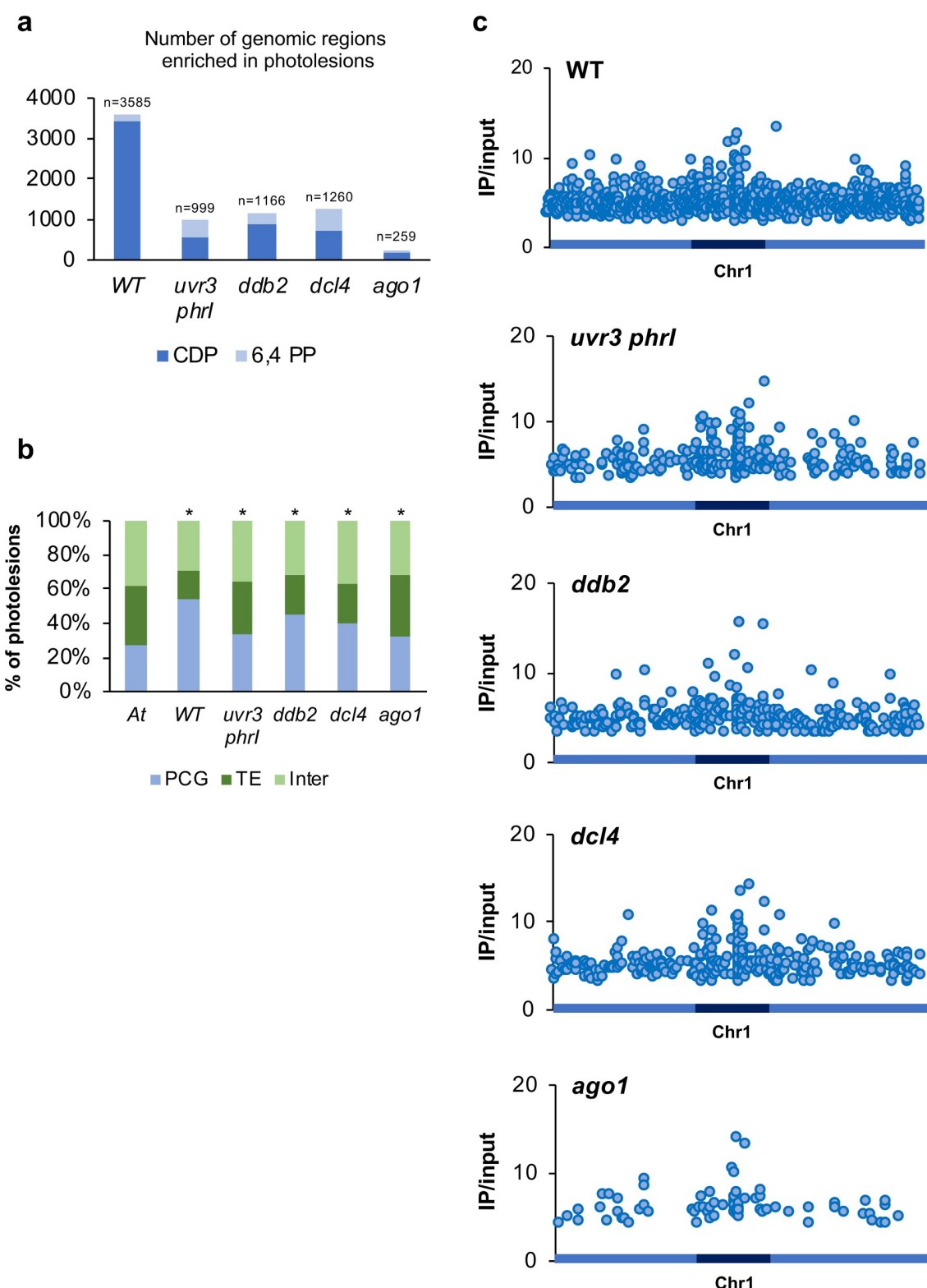

**Fig 5. Genome-wide identification of UV-C induced photolesions. a** Histograms representing the total number of photolesions identified in WT, *uvr3 phrI*, *ddb2*, *dcl4* and *ago1* using IPOUD. **b** Histograms representing the percentage of the identity (protein-coding genes: PCG, TE and intergenic) of photolesions identified in WT, *uvr3 phrI*, *ddb2*, *dcl4* and *ago1* plants. *A. t* represents the overall distribution in the Arabidopsis genome. * p< 0.01 compared to *A. t* according the chi2 test. **c** Distributions of photolesions along chromosome 1 (light blue: chromosome arms, dark blue: pericentromeric regions).

Interestingly, in all mutant plants, photolesions overlap with chromatin states 4, 5, 6, 8 and 9 with even a stronger enrichment compared to those of WT plants (S21 Fig). Thus, these observations highlight that photolesions formation rely on a complex interplay between genomic (di-pyrimidines) and epigenomic (chromatin states) features.

Given that small RNAs also contribute to the repair of UV-induced DNA lesions [16] we mapped the canonical siRNA populations (21-, 22- and 24-nt) to the photodamaged regions. Around 50% of the photolesions-containing regions overlap with small RNAs. They display POL IV dependency (S22 Fig) supporting the idea that these populations of siRNAs likely originated from POL IV precursors [16, 49]. Importantly, we could also observe that, in mutant plants, biogenesis of the 21-nt siRNA population does not fully rely on RNA POL IV (S22 Fig). These analyses highlight that the damaged loci are either under the transcriptional control of different types of RNA POL (i.e. RNA POL II and POL IV) or that defect in DNA repair processes may activate alternative small RNA biogenesis pathways [49].

Hence, these results suggest that, although the formation of photolesions depends primarily on the presence of di-pyrimidines, repressive chromatin states also influence the formation of photodamage. Moreover, our data confirm that both canonical and non-canonical siRNA biogenesis pathways interconnect at many damaged loci likely to contribute to maintain genome/methylome integrity.

## DNA methylation changes at UV-damaged sites

Upon induction of DNA damage and following DNA repair, epigenome integrity should be maintained by an accurate re-establishment of the pre-existing DNA methylation patterns. We aimed at determining at the genome-wide level whether regions that have undergone UV damage exhibit significant changes in their DNA methylation levels and how DNA repair pathways contribute to the maintenance of methylome integrity. In others words we are interested in determining if DNA damage could be a source of methylome changes.

For this, we compared the genomic location of photolesions with those of the identified DMRs. Using the cross-comparison of these genomic data we identified that 94 to 98.2% of the photodamaged regions did not exhibit methylome alterations (Fig 6A). Conversely, it implies that several genomic regions that have been UV-C-damaged exhibit significant DNA methylation changes. Indeed, the overlap between DNA damage and methylome change does not occur by chance, as supported by a representation factor >1 (Fig 6A). In WT plants, 0.9% of the damaged regions (33/3585) exhibited changes in DNA methylation (Fig 6A). This proportion exceeded 2.7% in all tested mutant plants, suggesting that specific DNA repair machinery efficiently coordinates re-establishment of DNA methylation upon repair (Fig 6A). In all plants, damaged genomic regions exhibiting DNA methylation changes, mapped predominantly to TE and to centromeric/pericentromeric regions, consistent with the enriched number of photolesions and DMRs identified in these parts of the genome (Figs 6B and S23).

In order to understand the role of the DNA methylation/demethylation pathways in the maintenance of methylome integrity upon UV exposure we extracted the DNA methylation level of each overlapping DMR from *met1*, *cmt2*, *cmt3* and *drm1/2* mutant plants [38]. For hyper-DMRs we found that, in all plants, the gain of methylation relies mainly on CMT2 and on the RdDM pathway, consistent with their roles in maintenance of CHH methylation and in *de novo* DNA methylation, respectively (Fig 7A, [18]). Interestingly, we also observed that, at few loci, CHH DNA methylation is also under the influence of MET1 in agreement with the ectopic gain of CHH methylation reported in *met1* plants [38] and the down regulation of *MET1* expression measured in WT, *ddb2* and *dcl4* plants (Figs 3A and S11). For hypo-DMRs the same trends could be observed with an additional role for CMT3 (S24A Fig) consistent

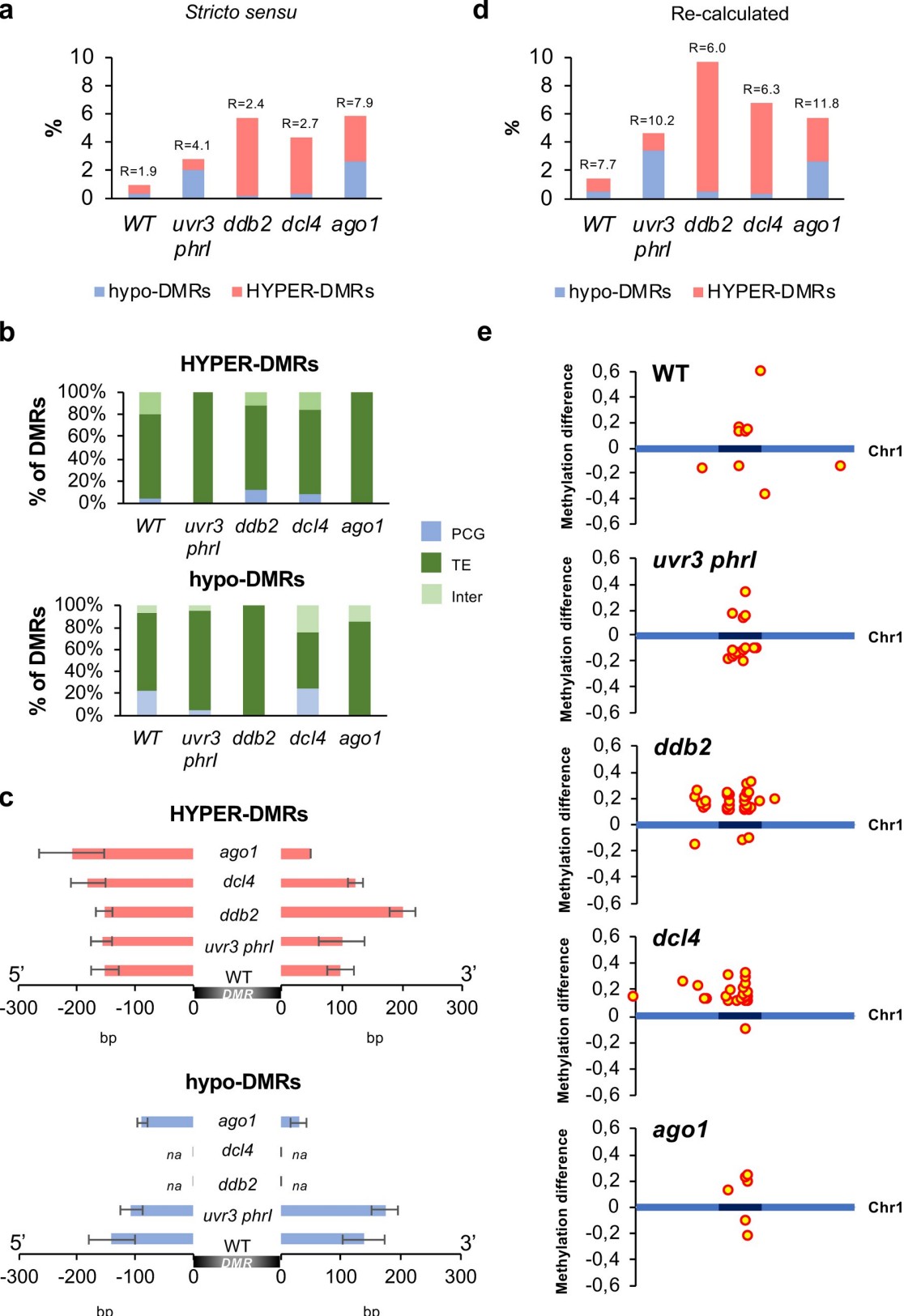

**Fig 6. DMRs overlapping with photolesions. a** Histograms representing the percentage of hypo- and hyper-DMRs overlapping with photolesions (*stricto sensu*) in WT, *uvr3 phrI*, *ddb2*, *dcl4* and *ago1* plants. R: Representation factor showing the statistical significance of the overlap between DMRs and photolesions (*stricto sensu*). **b** Histograms representing the percentage of the identity (protein-coding genes: PCG, TE and intergenic) of hyper- and hypo-DMRs overlapping with photolesions in WT, *uvr3 phrI*, *ddb2*, *dcl4* and *ago1* plants. **c** Histograms representing the average length (base pair: bp) of hypo- and hyper-DMRs spreading outside photolesions in WT, *uvr3 phrI*, *ddb2*, *dcl4* and *ago1* plants. na: non-applicable. **d** Histograms representing the percentage of the corrected hypo- and hyper-DMRs overlapping, with photolesions in WT, *uvr3 phrI*, *ddb2*, *dcl4* and *ago1* plants. R: Representation factor showing the statistical significance of the overlap between DMRs and photolesions. **e** Distributions of the corrected DMRs overlapping with photolesions along chromosome 1 (light blue: chromosome arms, dark blue: pericentromeric regions). Hyper- and hypo-DMRs are shown above and below each chromosome, respectively.

with the well-established link between CHG and CHH methylation [37]. Surprisingly, none of the damaged loci that have lost DNA methylation are target of the active DNA demethylation pathway, suggesting that defect in maintenance of DNA methylation may have predominantly led to hypo methylation (S24A Fig).

Importantly, genomic regions, such as TEs, can acquire DNA methylation through spreading from adjacent siRNA-targeted regions [52]. Given that 100% of the photodamaged regions exhibiting DMRs contain small RNAs, we investigated whether immediate adjacent regions gained or lost DNA methylation. Interestingly, we found that hyper-DMRs spread over 100 bp in 3' and 5' from the damaged sites in WT plants (Fig 6C). This holds true for all the mutant plants with a more pronounced effect in *ddb2* mutant plants (>200 bp in 3'; Fig 6C). We found the same trends for hypo-DMRs in WT, *uvr3 phrI* plants and to a lower extent in *ago1* plants (Fig 6C). In order to provide a more realistic view of the overlap between DNA damage and DMRs, we took into account this parameter and we recalculated the number of UV-damaged regions exhibiting DMRs. As expected, we observed that the proportion of photodamaged regions exhibiting methylation alterations increased for each plant (Fig 6D). Importantly, 75 to 100% of these regions are located within the centromeric/pericentromeric regions (Figs 6E, S25A and S25B).

To go further in the characterization of the UV-damaged regions exhibiting altered DNA methylation patterns, we analyzed their epigenomic features [36]. We realized that, in all plants, these genomic regions overlap predominantly with the chromatin state 5, enriched in the repressive H3K27me3 mark, representative of facultative heterochromatin and with the constitutive heterochromatic states 8 and 9 (S25C Fig). These observations are in agreement with the role of GGR in the repair of poorly transcribed or un-transcribed genomic regions [14]. In addition it highlights that DR likely contributes to the maintenance of methylome integrity in response to the formation of photolesions in silent genomic regions.

Given that RdDM likely plays an important role in the maintenance of proper DNA methylation level at photo-damaged sites, we determined the correlations between the different populations of canonical small RNAs (21-, 22- and 24-nt siRNA) mapping at these regions. Using circles of correlation, we found that for WT and *ddb2* hyper-DMRs, 21-nt and 22-nt populations positively correlated (Fig 7B). Conversely, 24-nt siRNA did not show linear correlation with the 2 other populations suggesting independent roles (Fig 7B). This holds true for hypo-DMRs in WT and *uvr3 phrI* plants (S24B Fig). In *dcl4* hyper-DMRs 22- and 24-nt small RNA positively correlated whilst 21-nt did not show linear correlation (Fig 7B). These observations suggest that in *dcl4* plants the canonical populations of small RNA may have different roles compared to WT and *ddb2* plants and that DCL4-independent processes exist to produce 21-nt (Fig 7B, [53]). Consistent with this, *dcl3dcl4*, *dcl2dcl3dcl4* and *nrpd1nrpe1* mutant plants displayed higher UV-C sensitivity than single mutant plants (S26A and S26B Fig), shedding light on synergism between 21-, 22- and 24-nt siRNA biogenesis/mode of action in response to UV-C irradiation. Hence, these analyses suggest that 21-, 22- and 24-nt siRNA likely act in

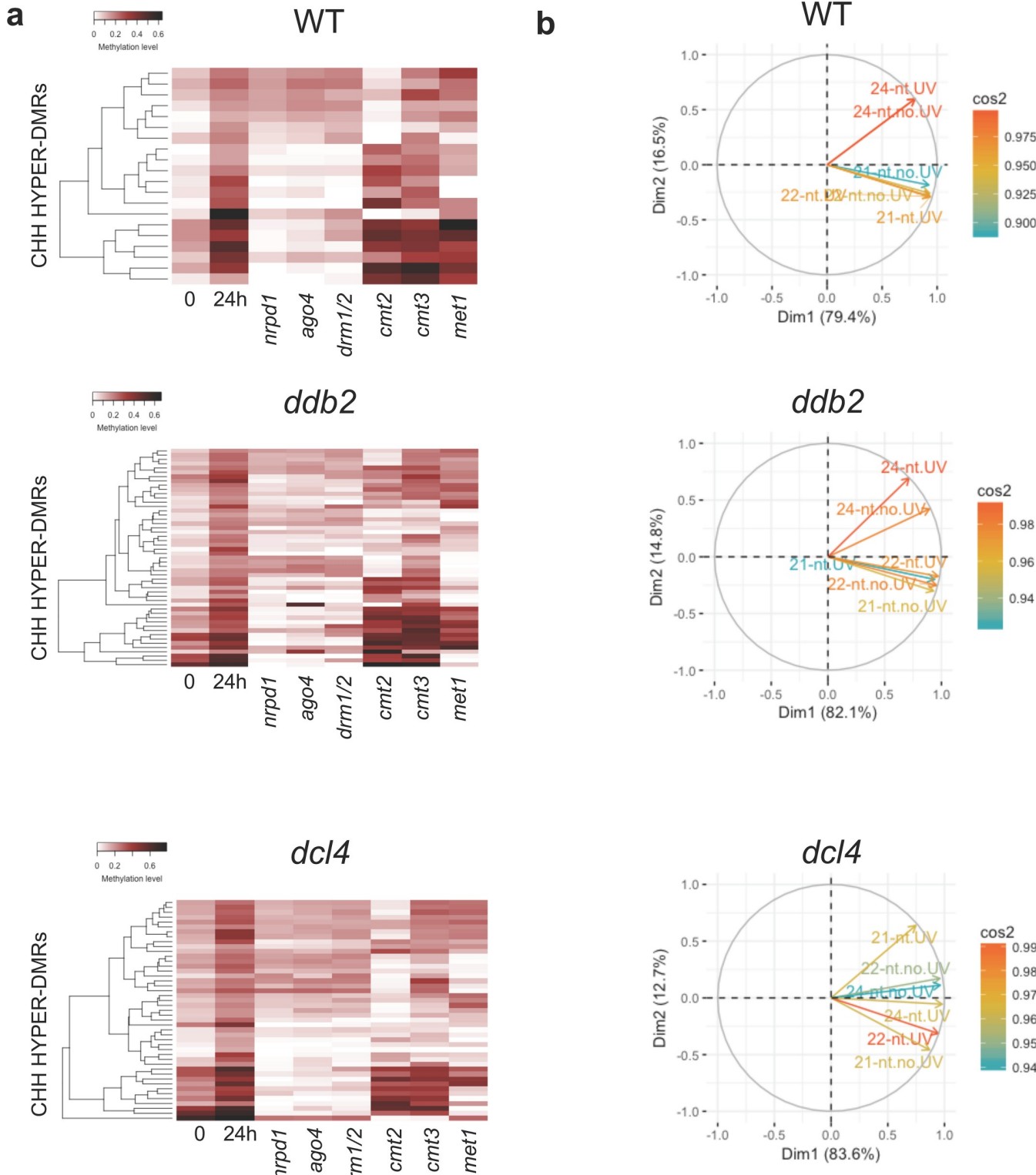

**Fig 7. Characteristics of hyper-DMRs overlapping with photolesions. a** Heatmaps of CHH methylation levels within hyper-DMRs identified in WT, *ddb2* and *dcl4* plants before, 24h upon UV-C exposure. Columns represent data for each indicated genotype (white, 0; black, 0.6) **b** Circles of correlations between 21-, 22- and 24-nt small RNAs mapping to the hyper-DMRs overlapping with photolesions in WT, *ddb2* and *dcl4* plants.

DNA repair and in DNA methylation, in agreement with the role of POL IV-dependent siRNA in DNA repair of UV-induced DNA lesions and in DNA methylation [16, 49].

Collectively, our data show that, in WT plants, UV-C irradiation leads to un-proper re-establishment of DNA methylation pattern in 1.5% of the damaged sites, which therefore correspond to the background level. The altered methylation profiles predominantly map to repressive chromatin and allows speculating that DNA damage are likely source of DNA methylation changes in silent chromatin. Moreover, these results revealed that a relationship exists between DNA damage/repair and methylome landscape with a strong contribution of the photodamage DNA repair pathways, including small RNA-mediated repair.

## Discussion

### UV-C irradiation induces DNA methylome alterations

Modifications of DNA methylation patterns have been characterized under different stress conditions and in several plant species [20]. Most of these studies were conducted using rather long exposure to biotic or abiotic stresses (several hours to few days) making difficult to determine a primary causal effect of the treatment on DNA methylation status [27, 28, 29, 30]. Here, we aimed at characterizing the effect of DNA damage/repair on methylome landscape. For this, we determined the DNA methylation profiles of *Arabidopsis thaliana* plants 24h upon few seconds of UV-C irradiation. In order to uncouple DNA damage/repair from light sensing/signaling [54] we used a short UV wave length (UV-C: 254 nm) that efficiently reacts with di-pyrimidines to form photodamage [3]. We found that UV-C irradiation, alters asymmetric DNA methylation landscape predominantly at centromeric and pericentromeric regions and that methylome changes are more pronounced in plants defective in the expression of specific repair factors of photolesions. Hence, we identified that DDB2, DCL4 and AGO1 prevent gain of DNA methylation and that UVR3-PHRI photolyases prevent loss of DNA methylation upon UV-C exposure.

DDB2 is involved in the recognition of photolesions during GGR [15, 40]. Although exclusively described as DNA repair factor, DDB2 was shown to form a complex with AGO4 to regulate the abundance of 24-nt siRNA at TE and repeats [23]. Indeed, in absence of applied stress, *ddb2* mutant plants displayed hundreds of hyper-DMRs [23]. Here we identified that *ddb2* mutant plants exposed to UV-C irradiation exhibit around 10,000 hyper-DMRs. Taken together these data show that DDB2 represses *de novo* DNA methylation during plant growth and also in response to UV exposure. In other words, it suggests that DDB2 may act as a general regulator of *de novo* DNA methylation during development and likely in response to stress. Recently, UVH6, a NER factor, was shown to be required for proper heat stress–induced transcriptional activation of heterochromatic TEs together with the mediator subunit MED14 [55]. These observations suggest that components of the DNA repair machinery play an important role in the regulation of silencing processes under stress conditions in addition to their canonical roles in DNA repair. These observations strengthen the notion that genome and methylome surveillance are coordinated.

Our methylome analyses revealed that *dcl4* and *ago1* deficient plants exhibit methylation changes closely related to those of *ddb2* plants. This supports the idea that both GGR and small RNA-mediated GGR cooperatively act in the maintenance of methylome integrity. Conversely to *ddb2*, *dcl4* and *ago1* plants, photolyases deficient plants mainly display hypo-DMRs upon UV-C exposure. In such plants the predominant repair pathway used is the NER, including GGR and small RNA-mediated GGR. Thus, the loss of DNA methylation may reflect the saturation of both GGR pathways including a strong reduction of the DDB2 contents due to its bias use. Upon induction of bulky DNA lesions, DDB2 is mobilized on chromatin to

recognize the DNA damage and is subsequently targeted to the 26S proteasome [15, 40, 56]. Such rapid DNA damage-dependent turnover suggests that DDB2 content is likely a limiting factor. Given that DDB2 also represses both ROS1 expression and activity [24], the UV-induced modulation of the DDB2 pool may lead to the misregulation of ROS1 content/activity in addition to AGO4 availability. Hence, DDB2 may fine tune the antagonist effect of active DNA demethylation and RdDM in response to UV exposure and therefore control methylome landscape at particular loci.

Importantly, our data show that specific DNA repair pathways, DNA synthesis-independent (DR) and -dependent (GGR and small RNA mediated-GGR), coordinate the efficient re-establishment of DNA methylation upon repair, linking the maintenance of genome and methylome integrities.

Interestingly, we identified that UV-C induced gain of CHH methylation as well as up-regulation of *CMT2* and *DRM2* expression, consistent with their roles in CHH DNA methylation [18]. Therefore, UV-C-induced deregulation of DNA methyltransferases may positively correlate with gain or loss of DNA methylation. Conversely, we observed that *MET1* expression was down-regulated in WT, *ddb2* and *dcl4* plants. Although CG methylation remained unaffected upon UV-C exposure, *met1* plants showed ectopic gain of CHH methylation [38]. This phenomenon could be due to UV-C release of expression of particular genomic regions (i.e. TE) followed by an expression-dependent gain of DNA methylation to trigger their silencing. In addition, we cannot exclude that DNA methyltransferases activity could have been modulated by UV-C treatment, even transiently.

The gain of DNA methylation observed in all plants predominantly relies on RdDM, consistent with its role in *de novo* DNA methylation [19]. Although, measurements of 24-nt siRNA abundance did not show significant increase 24h upon UV-C irradiation we cannot exclude that the 24-nt siRNA population was precociously and transiently modulated, reflecting a fast control of their biogenesis or of their mobilization into AGO4 to direct maintenance and/or *de novo* DNA methylation. Importantly the biogenesis of photoproduct-associated siRNAs involves a non-canonical pathway including RNA POL IV, RDR2 and DCL4 [16]. We identified that the biogenesis of 21-, 22- and 24-nt siRNA overlapping with the photodamaged regions predominantly relied on POL IV, in agreement with the hypothesis that such populations of siRNA originate from the same precursor and may serve for DNA repair and for DNA methylation [16].

Combination of post-transcriptional gene silencing (PTGS) and transcriptional gene silencing (TGS) processes acts to repress active plant retrotransposons [49, 57, 58]. Therefore, non-canonical siRNA biogenesis pathways and interconnections between RNA POL II and RNA POL IV biogenesis pathways may coexist at particular loci that need to be repaired and whose DNA methylation status needs to be tightly regulated [49]. This suggests that DR, GGR and small-RNA-mediated GGR coordinately act with the different DNA methylation pathways to maintain both genome and methylome integrities as a general (epi)genome immunity process [8,17].

## Photolesions are sources of DNA methylation changes in heterochromatin

In order to characterize the relationship between photoproducts locations and methylome changes, the genomic map of photolesions was produced using immunoprecipitation of both CPD and 6,4 PP followed by next generation sequencing. We found, in all tested plant, that photodamage were located genome wide, albeit a significant enrichment could be characterized in repressive chromatin, stressing the point that these genomic regions are more prone to form photoproducts. Interestingly, 5-mC adjacent to pyrimidine has higher absorbance in the UV-B range and is more prone to form pyrimidine dimers compared to the combination

with unmethylated cytosines [59]. Such feature is consistent with the higher enrichment of photodamage identified in constitutive heterochromatin where DNA methylation is concentrated. In addition, UV-C damages both euchromatin and heterochromatin whilst other genotoxic agents inducing bulky DNA lesions, like cisplatin, acts mostly in euchromatin [60]. Therefore, all these parameters allow considering that, in addition to the presence of di-pyrimidines, DNA methylation, nucleosome density and histone variants/PTM may synergistically contribute to favor photolesions formation in heterochromatin *vs* euchromatin. Collectively these studies highlight that heterochromatin likely displays higher genome/methylome flexibility than euchromatin and that DNA repair pathways (DR, GGR. . .) contribute to different extents to maintain its integrity.

We identified changes in CO and release of silencing of particular repeats consistent with heterochromatin reorganization observed during dark-light transition [61]. Interestingly, the photomorphogenic factor DE-ETIOLATED 1 (DET1) and DDB2 act together in the GGR pathway [62]. Thus, the light-induced chromocenters reorganization likely reflects the existence of a complex interplay between light signaling, induction/processing of photodamage and maintenance of methylome integrity.

The differential accessibility of euchromatic and heterochromatic regions for the DNA repair machinery implies that DNA repair is slower in compacted heterochromatin compared to relaxed euchromatin due to the kinetics of recruitment of the DNA repair factors to the damaged sites [63, 64]. Therefore, chromocenters reorganization upon UV-C exposure may reflect this slower repair kinetics and highlights that heterochromatic photodamage repair may require complex strategies as compared to euchromatic repair [64].

By comparing photolesions locations and DMRs we identified a significant overlap between DNA damage and methylation changes suggesting that photodamage are source of DNA methylation alterations. Interestingly, this un-proper re-establishment of methylome landscape is concentrated in repressed genomic regions (constitutive and facultative heterochromatin) and the DNA repair pathways prevent these exacerbated changes. The presence of TE and repeats in heterochromatin may explain such changes of DNA methylation levels in response to UV stress to further repress TE mobilization, genomic rearrangement that may affect genome integrity. Indeed, tight regulation of chromocenters compaction through specific epigenetic marks is important for chromosome architecture and segregation [65].

Importantly, we have to consider the effect of DNA damage on DNA methylation landscape may have been underestimated because UV-C irradiation also generates oxidatively induced DNA damage (i.e. 8-oxoguanine: 8-oxo-G) and DSB [10, 66]. Therefore, and similarly to photolesions, the proper re-establishment of DNA methylation profiles upon specific repair of these other types of damage could have also been affected.

UV-C exposure leads to the de-regulation of hundreds of genes, TE and repeats, within the first 24h hours following the treatment [35, 45, 48]. Heterochromatin-associated silencing was shown to be released in plants exposed to environmental cues, such as prolonged heat stress [48]. The temperature-induced release of silencing is transient, rapidly restored without the involvement of factors known to be required for silencing initiation [6]. Interestingly, we found that UV-C irradiation induced constitutive heterochromatin reshaping concomitantly with transcriptional reactivation of TE and repeats. Transposon bursts was reported in maize exposed to UV-B [44, 67] and light was shown to trigger changes in nuclear architecture including heterochromatin de-condensation [45, 48, 61]. Moreover, the *ONSEN* RNA steady state level increased upon UV-C exposure showing that such TE up-regulation reports both elevated temperature and high light stress, a characteristic of the global warming [43, 68, 69]. This reflects the complex interplays between environmental-induced transcriptional regulation, heterochromatin reorganization, DNA methylation and DNA damage-repair.

This study highlights that DNA repair pathways of UV-induced DNA lesions, namely DR, GGR and small-RNA mediated GGR, prevent excessive methylome alterations upon UV-C exposure. Moreover, we identified that photolesions are sources of DNA methylation changes in silent genomic regions suggesting that the efficiency of photodamage repair might play a significant role in the variation of DNA methylation landscapes likely contributing to reflect the evolutionary and life histories of plant species.

## Materials and methods

### Plant materials and growth conditions

The *Arabidopsis thaliana* wild-type (WT); *ddb2-3* [23], *uvr3 phrI* (WiscDsLox334H05 and WiscDsLox466C12, [39]), *nrpd1* (Salk_583051), *nrpe1* (Salk_029919), *dcl2-1* (Salk_064627), *dcl3-1* (Salk_005512), *dcl4-2* (GABI_160G05), *ago1-27* [42] plants used in this study are in the Columbia ecotype (Col0). Plants were grown *in vitro* on solid GM medium [MS salts (Duchefa), 1% sucrose, 0.8% Agar-agar ultrapure (Merck), pH 5.8] in a culture chamber under a 16 h light (light intensity $\sim$150 µmol m$^{-2}$ s$^{-1}$; 21˚C) and 8 h dark (19˚C) photoperiod.

### UV-C treatment

In order to prevent formation of photolesions, induced by the source of light, Arabidopsis WT and mutant plants were germinated and grown *in vitro* on solid GM medium for 10 days. Seedlings were subsequently transferred in larger Petri dishes (145 x 200 mm) at a density of 1 plant/cm and grown in the culture chamber for 11 additional days. Plants (40 plants/plate) were irradiated with UV-C (3,000 J/m$^2$) using Stratalinker. Immediately upon UV-C exposure 40–50 leaves from 2 different plates were harvested and pooled (time point 0). Remaining plants were put back in the growth chamber and 40–50 leaves from the same 2 different plates were harvested 24h upon UV-C exposure (time point 24h). Leaves samples of two biological replicates were pooled for genome wide studies.

### Immunoprecipitation of photodamaged DNA

Genomic DNA was extracted immediately upon UV-C exposure (time point 0) using the Plant DNA Extraction kit (Qiagen). Five µg of genomic DNA were sonicated (Diagenode Bioruptor: 18 x 30 s) and denatured 10 min at 95˚C in Buffer 1 (10 mM Tris HCl pH 7.5, 500 mM NaCl, 1 mM EDTA). Immunoprecipitation (IP) was performed by adding 5 µg of either anti-CPD mouse monoclonal antibody (CAC-NM-DND-001, Cosmo Bio, Japan) or anti-6,4 PPs (CAC-NM-DND-002, Cosmo Bio, Japan) and incubated on rotating wheel (8 rpm) overnight at 4˚C. Afterwards the suspension was incubated with M280 Dynabeads (Invitrogen) 4h at 4˚C under rotation (8 rpm). The pellet was washed 4 times with Buffer 1. The immunoprecipitated DNA was eluted with Buffer 2 (30 mM Tris HCl pH: 8.0; 150 µg Proteinase K) during 1h at 42˚C. DNA from the IP and input fractions was purified using the Nucleospin Gel and PCR clean-up kit (Macherey-Nagel). DNA from IP and input were used for library preparation and sequencing by Illumina Hi-Seq (paired-end 2*75 bp; FASTERIS, Switzerland). Sequences were mapped (input and IP) onto the Arabidopsis nuclear genome (TAIR10) using Bowtie 1.1.2 (-v2 –m1). Both CPD and 6,4 PP enriched regions were determined using MACS2 (version 2.1).

### Whole-genome bisulfite sequencing and mapping

Genomic DNA was prepared from the same set of plants used for photolesions immunoprecipitation (time point 0) and also from plants 24h upon UV-C exposure (time point 24h). Purified genomic DNA was bisulfite-treated and sequenced by Illumina Hi-Seq (paired-end 2*125

bp) by the FASTERIS Company (Switzerland). Conversion efficiency was determined using an unmethylated internal control (FASTERIS). For all samples conversion efficiency was >99.99%. Mapping on the arabidopsis genome (TAIR10) was performed using Bsmap (Bsmapz version) using default options (S2 Table).

### DMRs calling

Upon mapping, the methylation levels were calculated with methratio.py. Differentially Methylated Regions (DMRs) between untreated and treated plants of the same genotype were determined according Daccord et al. (2017) [70] considering 200 bp sliding-windows with an overlap of 50 bp (sliding-window-pipeline). DMRs were identified using the difference between identical windows and upon filtering. Consecutive windows exhibiting the same methylation change (gain or loss) were joined and methylation level recalculated. DMRs were called for a DNA methylation difference (p< 0.05 according Wilcoxon signed-rank test), within the same genotype, higher or equal than 0.4 for CG, 0.2 for CHG and 0.1 for CHH methylation contexts. We re-analyzed with our method previously published BS-seq datasets [38] for *met1* (GSM981031), *cmt2* (GSM981002), *cmt3* (GSM981003), *drm1/2* (GSM981015), *nrpd1* (GSM981039), *ago4* (GSM980991) and *ago6* (GSM980993). Methylation levels were also calculated from chosen genomic coordinates.

The overlaps between the characterized DMRs, the photolesions enriched genomic regions and the chromatin states [36] were performed using the web assisted tool (https://usegalaxy.org/; "Operate on genomic intervals") with an overlap size of 50%.

### Small RNA sequencing

Small RNAs were prepared from the untreated and UV-C treated plants (time points 0 and 24h) using the Tri-Reagent (Sigma), used for library preparation and sequencing by Illumina Hi-Seq (single end 50 bp; FASTERIS, Switzerland). Reads were aligned and mapped onto the Arabidopsis genome (TAIR10) using Bowtie (version 1.2.1.1; parameters: -y -e 50 -n 0 -a—best—strata –nomaqround; S2 Table). Upon conversion with samtools (version 1.5), reads overlapping with either the UV-damaged loci or the DMRs (50% overlap) were calculated with intersectBed (BED tools version 2.27.1). The read counts are divided by the "per million" scaling factor (RPM).

### RT-qPCR

Reverse transcription (RT) was performed on total RNA extracted using Tri-Reagent (Sigma) from untreated and UV-C treated plants (time points 0, 2h and 24h). The RT reaction was performed on 5 µg of total RNA using a mixture of random hexamers-oligo d(T) primers and the cDNA reverse transcription kit (Applied Biosystems). 100 ng of the RT reaction was used for quantitative PCR (qPCR). qPCR was performed, including technical triplicates, using a Light Cycler 480 and Light Cycler 480 SYBR green I Master mix (Roche) following manufacturer's instructions. All primers are listed in Supplemental S3 Table. Experiments were at least duplicated using independent biological replicates.

### UV-C root growth assay

UV-C sensitivity was performed using 7-day-old *in vitro* germinated WT and mutant plants. Plants were grown vertically on square plates containing GM medium. Root length was measured 24h upon UV-C exposure (900 J/m$^2$) using the Stratalinker 2400 (Stratagene). The relative root growth was calculated: (root length treated/ root length untreated) ×100 (±SD). Eight plants per replicate were used. Experiments were performed in triplicates.

## Tissue fixation and 5-mC immunolocalization

Leaves numbers 3 and 4 of 21-old day Arabidopsis WT and mutant plants were collected before irradiation (time point 0) and 24h upon UV-C exposure as described above. The collected leaves were fixed in 4 successive washing steps of at least 5 min in fixative solution (3:1 ethanol / acetic acid) and stored at -20˚C. Fixed leaves were washed twice with demineralized water and incubated 3h at 37˚C in a digestion mix (0.3% cellulase, 0.3% pectolyase 10 mM Na-citrate pH = 4.5). Digested leaves were spread on poly-lysine slides using 20 μl of acetic acid solution (60%) at 46˚C for 1 min. Slides were washed 3 times in fixative solution and once in sterilized demineralized water. Post fixation was performed in a 2% paraformaldehyde PBS solution for 5 min. Slides were washed with demineralized water and incubated for 1h at room temperature in permeabilization buffer (8% BSA, 0.01% Triton-X in Phosphate Buffer Saline x1). For 5-mC immunolocalization slides were first incubated over night at 4˚C with a monoclonal anti-5-mC-antibody (Diagenode C15200003; 1/1000 dilution in 1% BSA, Phosphate Buffer Saline x1). Upon incubation slides were washed 3 times with PBS and goat anti-mouse antibody coupled to FluoProbes488 (Interchim FP-GAMOTTGO488; 1/200 dilution in 1% BSA, Phosphate Buffer Saline x1) was added for 90 min at room temperature. Finally, slides were washed 3 times with Phosphate Buffer Saline x1 and 15 μl of Fluoromount-G (Southern Biotechnology CAT NO 0100–01) with 2 μg/ml DAPI were added as mounting solution for the coverslip.

## Image quantification

Whole image acquisition was performed on a Zeiss LSM 780 confocal microscope using a 64X oil immersion objective. 405 nm and 488 nm laser excitation wavelengths were used for DAPI and for the FluoProbes488, respectively. DAPI emission was measured between 410 nm and 585 nm wavelength on a first track. FluoProbes488 emission was measured between 493 nm and 630 nm wavelength on a second track. The same acquisition gain settings were used for all slides of a same genotype. Slight adjustment was performed due to labeling differences in between experiences. Acquisition gain for DAPI and FluoProbes488 in WT, *uvr3 phrI*, *ddb2*, *dcl4* and *ago1* were [650, 500], [750, 698], [800, 615], [750, 698] et [800, 615] respectively. Each image acquisition consists in a Z-stack capture with a 0.43 μm slice distance. All pictures had a final voxel size of 0.1014 x 0.1014 x 0.4300 μm^3.

The image quantification was performed on ImageJ1.52o using a homemade plugin called Nucl.Eye (https://github.com/mutterer/Nucl.Eye). This plugin allows either an automatic or a manual delimitation of all nuclei on the z-max compiled image in order to quantify size and signal intensity of each nuclei and all internal spot like entities. Using the information of individual spot (intensity, surface) and nuclei surface, the Chromocenter Occupancy (CO) was defined as percentage of surface occupied by all bright DAPI spots (chromocenters) in the corresponding nucleus. Intensities and surfaces of each chromocenter and those of the whole nucleus were used to calculate their relative intensities and surface.

## Statistics

Mann-Whitney U or Wilcoxon Matched-Pairs Signed-Ranks tests were used as non-parametric statistical hypothesis tests (http://astatsa.com/WilcoxonTest/). Chi 2 test was used to determine significant difference between categories distribution (https://goodcalculators.com/chi-square-calculator/). Representation factor (R) was used to determine the statistical significance of the overlap between 2 independent groups of genomic regions (http://nemates.org/MA/progs/overlap_stats.html). t-test was used as parametric statistical hypothesis test.

## Supporting information

**S1 Fig. DMRs and chromatin states. a** Histograms representing the percentage of hypo-DMRs (red) and hyper-DMRs (blue) identified in WT, *uvr3 phrI*, *ddb2*, *dcl4* and *ago1* plants 24h upon UV-C exposure.

**b** Histograms representing the distribution of the chromatin states overlapping with DMRs identified in WT, *uvr3 phrI*, *ddb2*, *dcl4* and *ago1* plants. *A. t* represents the overall distribution of the 9 chromatin states in the Arabidopsis genome (*A. t*). * Chi square test < 0.01 compared to the Arabidopsis genome.

**c** Histograms representing the distribution of TE families overlapping with DMRs identified in WT, *uvr3 phrI*, *ddb2*, *dcl4* and *ago1* plants. *A. t* represents the overall distribution of the TE families in the Arabidopsis genome (*A. t*). * Chi square test < 0.01. compared to the Arabidopsis genome.
(TIFF)

**S2 Fig. Genome wide distribution of DMRs and methylation changes. a** Circos representation of the DMRs identified in WT, *uvr3 phrI*, *ddb2*, *dcl4* and *ago1* plants 24h upon UV-C exposure.

**b** Boxplots representing the CHH methylation changes in WT, *uvr3 phrI*, *ddb2*, *dcl4* and *ago1* plants 24h upon UV-C exposure.
(TIFF)

**S3 Fig. DNA methylation levels.** CHH DNA methylation levels along chromosomes(light grey: chromosome arms, dark gray: pericentromeric regions) in WT, *uvr3 phrI*, *ddb2*, *dcl4* and *ago1* plants prior (0) and 24h upon UV-C exposure.
(TIFF)

**S4 Fig. DMRs features. a** Boxplots representing the size (bp) of hyper-DMRs (left panel) and hypo-DMRs (right panel) identified in WT, *uvr3 phrI*, *ddb2*, *dcl4* and *ago1* plants. Exact p values according Mann Whitney test are indicated above each graph.

**b** Genome browser views of hyper- and hypo-DMRs identified in WT, *uvr3 phrI*, *ddb2*, *dcl4* and *ago1* plants upon UV-C exposure. Upper panel: Chr1: 15, 023, 000 bp-15, 027, 000 bp intergenic region. Lower panel: Chr5: 20, 127, 000 bp-20, 130, 000 bp intergenic region/PCG. Red line: HYPER-DMRs; blue line: hyo-DMRs.

**c** Histograms representing the distribution of methylation levels prior UV-C irradiation in genomic regions exhibiting hypo- (left panel) and hyper-DMRs (right panel).

**d** Boxplots representing the CHH methylation levels prior UV-C irradiation in genomic regions exhibiting hypo- (left panel) and hyper-DMRs (right panel) identified in WT, *uvr3 phrI*, *ddb2*, *dcl4* and *ago1* plants. Exact p values according Mann Whitney test are indicated above each graph.

**e** Boxplots representing the CHH methylation difference of hypo- (left panel) and hyper-DMRs (right panel) identified in WT, *uvr3 phrI*, *ddb2*, *dcl4* and *ago1* plants 24h upon UV-C exposure. Exact p values according Mann Whitney test are indicated above each graph.
(TIFF)

**S5 Fig. *ONSEN* RNA level and DMRs. a** Genome browser showing the density graph 24-nt siRNA abundance and DMRs at *ONSEN* locus in WT, *uvr3 phrI*, *ddb2*, *dcl4* and *ago1* plants prior UV-C irradiation (0) and upon UV-C exposure (24h).

**b** RNA steady state level of *ONSEN* transcripts determined by RT-qPCR in WT, *uvr3 phrI*, *ddb2*, *dcl4* and *ago1* plants before (0), 2h and 24h following UV-C irradiation.
(TIFF)

**S6 Fig. Genome browser views.** Examples of hyper- and hypo-DMRs identified in WT, *uvr3 phrI*, *ddb2*, *dcl4* and *ago1* plants upon UV-C exposure in TE enriched region (**a**) and in intergenic regions surrounding protein coding genes (**b**) or TE (**c**).
(TIFF)

**S7 Fig. Overlap between hyper-DMRs.** Venn diagrams representing the overlap of hyper-DMRs between WT plants and either *uvr3 phrI*, *ddb2*, *dcl4*, *ago1* or WT drought plants as well as in between mutant plants. R: Representation factor and exact p value showing the statistical significance of the overlap.
(TIFF)

**S8 Fig. Overlap between hypo-DMRs.** Venn diagrams representing the overlap of hypo-DMRs between WT plants and either *uvr3 phrI*, *ddb2*, *dcl4*, *ago1* or WT drought plants as well as in between mutant plants. R: Representation factor and exact p value showing the statistical significance of the overlap.
(TIFF)

**S9 Fig. Comparisons of DNA methylation levels.** Boxplot of CHH methylation levels within DMRs identified in WT, *uvr3 phrI*, *ddb2*, *dcl4* and *ago1* plants 24h upon UV-C exposure. The CHH methylation levels of each of these DMRs are reported before and upon UV-C exposure for each plant. Exact p values according Wilcoxon signed rank test are indicated above each graph. Blue significant decrease of DNA methylation level, red significant increase of DNA methylation level, black non-significant change.
(TIFF)

**S10 Fig. Relative fluorescence intensities of chromocenters, chromocenters and nuclei surfaces.** Boxplots representing the relative fluorescence intensities of chromocenters (**a**), chromocenter surface (**b**) and nucleus surface (**c**) of untreated (time point 0) and UV-C treated (time point 24h) WT, *uvr3 phrI*, *ddb2*, *dcl4* and *ago1* plants. Exact p values according Mann Whitney test are indicated above each graph. Number of chromocenters analyzed: 252 to 601; Number of nuclei analyzed: 44 to 109.
(TIFF)

**S11 Fig. *180 bp* and *5S* RNA levels.** Relative RNA steady state level (±SD) of *180 bp* (**a**) and *5S* RNA (**b**) transcripts determined by RT-qPCR in WT, *uvr3 phrI*, *ddb2*, *dcl4* and *ago1* plants before (0), 2h and 24h following UV-C irradiation.
(TIFF)

**S12 Fig. DNA methyltransferases RNA levels.** Relative RNA steady state level (±SD) of *MET1*, *CMT2*, *CMT3* and *DRM2* transcripts determined by RT-qPCR in WT, *uvr3 phrI*, *ddb2*, *dcl4* and *ago1* plants before (0), 2h and 24h following UV-C irradiation.
(TIFF)

**S13 Fig. DNA demethylases RNA levels.** Relative RNA steady state level (±SD) of *ROS1*, *DML2* and *DML3* transcripts determined by RT-qPCR in WT, *uvr3 phrI*, *ddb2*, *dcl4* and *ago1* plants before (0), 2h and 24h following UV-C irradiation.
(TIFF)

**S14 Fig. RNA-directed DNA methylation and hypo-DMRs. a** Heatmaps of CHH methylation levels within hypo-DMRs identified in WT, *uvr3 phrI*, *ddb2*, *dcl4* and *ago1* plants before, 24h upon UV-C exposure. The CHH methylation levels of each of these hypo-DMR are reported for RNA POL IV (*nrpd1*) and AGO4 (*ago4*) deficient plants. (white: 0; black: 1).
**b** Boxplots representing the abundance of 24-nt siRNAs mapping to the CHH hypo-DMRs

identified in WT, *uvr3 phrI*, *ddb2*, *dcl4*, *ago1* plants for protein-coding genes (PCG), TE and intergenic regions. For each genotype the abundance of 24-nt siRNAs is shown in RNA POL IV deficient plants (*nrpd1*). The 24-nt siRNA abundance is normalized against global small RNA content and expressed as reads per million (RPM). p-values are calculated according to Wilcoxon Matched-Pairs Signed-Ranks.
(TIFF)

**S15 Fig. DMR, active DNA demethylation and 24-nt siRNA. a** Histograms representing the percentages of hypo-DMRs identified in WT, *uvr3 phrI*, *ddb2*, *dcl4*, *ago1* plants overlapping with *rdd* hyper-DMRs.
**b** Boxplots representing the global 24-nt siRNA abundance before and 30 min following UV-C exposure. p-value is calculated according to Wilcoxon Matched-Pairs Signed-Ranks.
(TIFF)

**S16 Fig. 21-nt, 22-nt siRNA abundances at hyper-DMRs.** Boxplots representing the abundance of 21-nt **(a)** and 22-nt **(b)** siRNAs mapping to the CHH hyper-DMRs identified in WT, *uvr3 phrI*, *ddb2*, *dcl4*, *ago1* plants for protein-coding genes (PCG), TE and intergenic regions. For each genotype the abundance of 21-nt and 22-nt siRNAs is shown in RNA POL IV deficient plants (*nrpd1*). The 21-nt and 22-nt siRNA abundance are normalized against global small RNA content and expressed as reads per million (RPM). p-values are calculated according to Wilcoxon Matched-Pairs Signed-Ranks.
(TIFF)

**S17 Fig. 21-nt, 22-nt siRNA abundances at hypo-DMRs.** Boxplots representing the abundance of 21-nt **(a)** and 22-nt **(b)** siRNAs mapping to the CHH hypo-DMRs identified in WT, *uvr3 phrI*, *ddb2*, *dcl4*, *ago1* plants for protein-coding genes (PCG), TE and intergenic regions. For each genotype the abundance of 21-nt and 22-nt siRNAs is shown in RNA POL IV deficient plants (*nrpd1*). The 21-nt and 22-nt siRNA abundance are normalized against global small RNA content and expressed as reads per million (RPM). p-values are calculated according to Wilcoxon Matched-Pairs Signed-Ranks.
(TIFF)

**S18 Fig. DMRs in *ago4* and *ago6*.** Heatmaps of CHH methylation levels within hyper- **(a)** and hypo-DMRs **(b)** identified in WT, *uvr3 phrI*, *ddb2*, *dcl4* and *ago1* plants before and 24h upon UV-C exposure. The CHH methylation levels of each of these DMRs are reported for AGO4 (*ago4*) and AGO6 (*ago6*) deficient plants. Columns represent data for each indicated genotype (white: 0; black: 1).
(TIFF)

**S19 Fig. Photolesions location.** Circos representation of photolesions identified in WT, *uvr3 phrI*, *ddb2*, *dcl4* and *ago1* plants.
(TIFF)

**S20 Fig. Di-pyrimidines frequencies at photolesions.** Boxplots representing the di-pyrimidines frequencies (CC, TT, TC and CT) for each DNA strand (+ and–strand) in photodamaged regions (intergenic, TE and protein-coding genes: PCG) identified in WT, *uvr3 phrI*, *ddb2*, *dcl4* and *ago1* plants. The frequency of di-pyrimidine in the *Arabidopsis thaliana* (*A. t*) genome is also represented.
(TIFF)

**S21 Fig. Photolesions and chromatin states.** Boxplots representing the chromatin states overlapping with photolesions enriched regions in WT, *uvr3 phrI*, *ddb2*, *dcl4* and *ago1*

plants.
(TIFF)

**S22 Fig. siRNA overlapping photodamaged regions.** Boxplots representing the abundance of 21-, 22 and 24-nt siRNAs mapping to the photodamaged genomic regions in WT, *uvr3 phrI*, *ddb2*, *dcl4*, *ago1* plants. For each genotype the abundance of 21-, 22 and 24-nt siRNAs is shown in RNA POL IV deficient plants (*nrpd1*). siRNA abundances are normalized against global small RNA content and expressed as reads per million (RPM). p-values are calculated according to Wilcoxon Matched-Pairs Signed-Ranks.
(TIFF)

**S23 Fig. Photolesions and DMRs.** Distributions of DMRs overlapping (*Stricto sensu*) with photolesions along the Arabidopsis chromosomes (light blue: chromosome arms, dark blue: pericentromeric regions). Hyper- and hypo-DMRs are shown above and below each chromosome, respectively.
(TIFF)

**S24 Fig. Characteristics of hypo-DMRs overlapping photolesions. a** Heatmaps of CHH methylation levels within hypo-DMRs identified in WT and *uvr3 phrI* plants before, 24h upon UV-C exposure. Columns represent data for each indicated genotype (white, 0; black, 0.6).
**b** Circles of correlations between 21-, 22- and 24-nt small RNAs mapping to the hypo-DMRs overlapping with photolesions in WT and *uvr3 phrI* plants.
(TIFF)

**S25 Fig. DMRs overlapping photolesions. a** Distributions of DMRs overlapping with photolesions along the arabidopsis chromosomes (light blue: chromosome arms, dark blue: pericentromeric regions). Hyper- and hypo-DMRs are shown above and below each chromosome, respectively.
**b** Histogram representing the percentage of hyper- and hypo-DMRs overlapping with photolesions located within centromeric and pericentromeric regions.
**c** Histograms representing the distribution of the 9 chromatin states of DMRs overlapping with photolesions.
(TIFF)

**S26 Fig. UV sensitivity of DCLs and RNA POL IV/V loss of function arabidopsis plants. a** Genetic interaction between *dcl2, dcl3 and dcl4*. Seven-day-old WT, single (*dcl2, dcl3 and dcl4*) and double (*dcl2/3, dcl2/4, dcl3/4 and dcl2/3/4*) mutant plants were exposed to UV-C (900 J/m$^2$). Root growth was calculated relative to the corresponding untreated plants (±SD). Eight plants per replicate were used and experiments were triplicated. t-test *p<0.01 compared to WT; ** p<0.01 compared to *dcl3 and dcl4*; ns: non-significant.
**b** Genetic interactions between *nrpd1* and *nrpe1*. Seven-day-old WT, single (*nrpd1* and *nrpe1*) and double (*nrpd1nrpe1*) mutant plants were exposed to UV-C (900 J/m$^2$). t-test *p<0.01 compared to WT; ** p<0.01 compared to each single mutant.
(TIFF)

**S1 Table. Statistics of di-pyrimidines frequencies.**
(DOCX)

**S2 Table. NGS statistics.**
(DOCX)

**S3 Table. qPCR primers.**
(DOCX)

## Acknowledgments

We thank Amina Mehidi for her help in statistical analyses and members of the Schmit-Chabouté's group.

## Author Contributions

**Formal analysis:** Stéfanie Graindorge, Valérie Cognat, Philippe Johann to Berens, Jérôme Mutterer, Jean Molinier.

**Funding acquisition:** Jean Molinier.

**Investigation:** Jean Molinier.

**Resources:** Stéfanie Graindorge, Valérie Cognat, Jérôme Mutterer.

**Software:** Jérôme Mutterer.

**Supervision:** Jean Molinier.

**Validation:** Jean Molinier.

**Writing – original draft:** Philippe Johann to Berens, Jean Molinier.

**Writing – review & editing:** Jean Molinier.

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
