## [Decision Letter · Decision Letter 0]

7 Aug 2019

Dear Dr Molinier,

Thank you very much for submitting your Research Article entitled 'Photodamage repair pathways contribute to the accurate maintenance of the DNA methylome landscape upon UV exposure' to PLOS Genetics. Your manuscript was fully evaluated at the editorial level and by independent peer reviewers. While reviewers 1 and 3 were generally supportive of the manuscript, reviewer 2 raised some detailed, and we believe, very important criticisms of the data and its ability to support the conclusion that you draw. Based on the reviews, we will not be able to accept this version of the manuscript, but we would be willing to review again a much-revised version, in which you address the points raised by the reviewers or explain satisfactorily why they are invalid. We cannot, of course, promise publication at that time.

If you decide to revise the manuscript for further consideration at PLOS Genetics, please aim to resubmit within the next 60 days, unless it will take extra time to address the concerns of the reviewers, in which case we would appreciate an expected resubmission date by email to plosgenetics@plos.org.

[LINK]

We are sorry that we cannot be more positive about your manuscript at this stage. Please do not hesitate to contact us if you have any concerns or questions.

Yours sincerely,

Julian E. Sale

Associate Editor

PLOS Genetics

Wolf Reik

Section Editor: Epigenetics

PLOS Genetics

Reviewer's Responses to Questions

**Comments to the Authors:**

Reviewer #1:

General comment: 

This manuscript described the sophisiticated interconnections between maintenance of genome and methylome integrities in response to UV-C irradiation in Arabidopsis thaliana. Herein, the authors reported that (1) UV-C irradiation leads to alterations of DNA methylation in heterochromatic regions; (2) Photodamage repair pathways prevent excessive changes of methylome landscape; (3) Constitutive heterochromatin organization is strongly affected upon UV-C exposure; and (4) Photodamage are sources of DNA methylation changes. The above findings add a new level of knowledge in the complex interplays between DNA damage, DNA repair and DNA methylation dynamics. The research content of the manuscript is suitable for the fields that of the PLOS Genetics focus on. 

Specific comments: 

The English in the manuscript is good basically, however it is still necessary to check and revise for some sentences and words carefully, some suggestions are as below for authors’ references. 

Line 307, for “UV-C irradiation induces chromocenters reshaping” , “induces” is better than “induced”. Line 215 et al. 

In text, authors use Fig, but for lines 1098-1170, authors use Figure. 

Lines 193 and 195, Figs are better. 

Line 741, materials are better. 

Lines 800 and 811, the untreated is better than untreated.

Reviewer #2: This study examines interactions between DNA repair in response to UV and maintenance of DNA methylation. This is an exciting interaction about which we know little. While the premise of the paper is interesting on the whole it is confusingly written and presented. The hypotheses being tested, particularly in reference to mutants, are unclear. The statistical and bioinformatics analyses are described insufficiently. There are exciting parts to the paper - for example direct mapping of photolesions via ChIP-seq and BS-seq following UVC exposure - but the results and approaches are very poorly described making it hard to be confident in the conclusions reached.

UV exposure in plants is known to generate CPD and 6,4PP lesions between di-pyridimines. These can be repaired by different pathways; first ‘direct repair’ via DNA photolyases, which in Arabidopsis are PHR1 and UVR3, alternatively NER can act which the authors divide into TCR (CSA and CSB dependent) and GGR (DDB2 dependent) pathways. Interestingly DDB2 has been shown to interact with AGO1 and bind siRNAs that originate from photodamaged regions.

The authors perform BS-seq of Arabidopsis after UV-C exposure. This identifies 2,000 DMRs, half of which are hyper and half are hypo. How many replicates were performed for this and how consistent was DMR identification between replicates? Presumably independent UV-C treatment would induce variable patterns of damage caused by stochastic exposure of different genome regions to radiation in different cells? At line 192 - are these DMRs changed in all sequence contexts, or just CG, CHG or CHH? Lines 194-197 - this is a little confusing - H3K27me3 is typically not found in DNA methylated regions (at least in wild type), so its surprising to see this overlap reported. Could the authors provide more detail on how overlap analysis was performed? Line 199-200 I am very unclear with what the authors mean by constitutive versus facultative heterochromatin - please explain this more clearly and show how overlap was assessed here. Line 197 - Im also not clear what this statement about chromatin states and their overlap differing significantly from the genome average means - please explain. If the DMRs are found more frequently in heterochromatin does this not simply arise from the fact that heterochromatin has higher methylation levels?

In Figure 1 no information is presented on the reproducibility of these results - for each genotype how many replicates were performed and what correlation was observed between the changes? Aside from plotting DMRs as points in 1B could the authors also simply plot the moving average of CG CHG and CHH methylation along the chromosome? To what extent are total methylation levels changed? One might expect that CHH levels would show some difference from this way of looking?

Line 232 onwards. The authors describe DMRs in several genotypes - what is the overlap between these DMRs between genotypes? Again replicate information is lacking. On line 240 the authors mention that DDB2 has been connected with AGO4 - were ago4 mutants also tested in addition to ago1? Line 241-242 I don't agree with this sentence - how do the detection of DMRs in ddb2 support a role for small RNAs? The authors again mention DMR overlap with TEs - the authors need to perform a test whether this is significantly more than expected, especially as TEs and methylation and strongly correlated in the first instance. The size of DMRs is described as being changed in the mutants compared to wild type - please state what the difference is in the text - is this change significant? The authors propose that this reflects a change in spreading - this seems poorly supported by the presented data. It would be useful if the authors could provide screenshots from browsers (including replicate samples) showing clear examples of DMRs in each case, in addition to examples showing the stated change in spreading of methylation.

As phenotypes are observed with dcl4, have the authors also tested rdr6, which shares phenotypes and siRNA changes with dcl4?

In Figure 2a could the authors provide a clearer example of what ‘% chromocenter occupancy’ represents? The images in part b do not present a very convincing case for meaningful changes. Based on the images shown it is very hard to conclude that there are any changes here. Perhaps measurement in mutants such as ddm1 or met1 would yield a more convincing change to calibrate against?

Line 201. Please explain more clearly how the conclusion that CHH sites are most changed has been reached?

Is the 97% referring to DMRs, or to specific cytosines that have been identified as changed. Lines 298-300: CHH patterns between several of the mutants analysed are described as ‘rather resembling’ - what does this mean? Please try and be more quantitative and rigorous in your analysis and description of results.

Line 213. Again Im very confused about what the authors are defining as facultative versus constitutive heterochromatin? This distinction is made throughout the paper with no explanation of how these sequences are defined.

Line 264-265. What does this statement mean? How are the TEs in question ‘over-represented compared to their distribution’ - please provide a clearer explanation of how this is tested for. Also Class I TEs encompass more types of TE than just LTR/Gypsy. Do the authors mean all RNA TEs here, or just Gypsy? Again it would be beneficial to the reader to have the BS-seq data presented in a browser at loci of interest that represent clearly the types of changes reported via genome-wide analyses. The authors invoke UV-C induced changes to DNA methylation as a TE defense mechanism. However, no attempt is made to test for increased transposition after UV-C exposure in the mutants tested - this should be performed if they want to propose this.

The authors continue to analyse siRNA patterns around the identified DMRs and observe various correlations. I am not clear how these relate to the original hypothesis? Have the authors sequenced siRNAs during the UV-C response? The desicriptions used are also imprecise and opaque in meaning at many points - for example line 428 ‘siRNA patterns look much more complex’ - what does this mean? How is complexity assessed?

An interesting approach is made to map DNA damage by using ChIP with CPD and 6,4-PP antibodies. However, again the authors provide no information on replicates and how consistent the patterns observed are between independent replicates. Is it possible to repeat this experiment using different UV exposures to confirm that the signal increases with higher levels of damage? On lines 453-454 - what mutants are being referred to here? Please be more specific? Which mutants were tested and why? Why would photolesion enrichment go down in these mutants (whichever they are?)? Lines 456-458 - this is very interesting potentially - so TEs/intergenic regions are more prone to damage? This could represent (i) higher dimer formation in heterochromatin, (ii) reduced repair in heterochromatin, (iii) increased repair in euchromatin - or all/a combination of these factors - which do the authors think is occuring? Line 459 - the authors state here that they think that heterochromatin has ‘high reactivity’ please clarify your thoughts here? What does higher reactivity mean at the molecular level? Would you expect any of your mutants tested to change this? It would be helpful to see a browser track showing the ChIP DNA samples (and replicates) in enriched versus non-enriched regions? Perhaps a region that contains both expressed genes and heterochromatic TEs - can the difference in signal described be observed in such a way? Lines 467-470 - please explain more about these chromatin states - what are they? Which marks are they typified by? What biological roles do they play in the genome? Lines 470 - all mutants are said to ‘behave like WT’ - please explain - you mean the lesion enriched sites overlap the same places? (although they are fewer?) - but with a ‘more pronounced effect’ - so they are not the same then? Please clarify your meaning here.

Generally the reproducbility of the approach is an interesting question. At Line 499 the authors compare photolesion ChIP-seq data with the loci that lose methylation after UV-C exposure. Were the photolesion and BS-seq data performed on the same tissue? This again comes back to replication - how many times has each experiment been repeated and how reproducible are the signals? Lines 504-505 1% of damaged regions show a change in DNA methylation (numerically how many places are being referred to here?). It is stated that the ‘DNA repair defective’ plants show a 2% change - is this significant? Which defective mutants are specifically being referred to?

Minor comments:

Abstract line 29 - please clarify the use of ‘asymmetric’ here - do you mean CHH sites?

Abstract line 38 - please provide a more explicit definition of constitutive and facultative heterochromatin.

Line 125 - Perhaps add ‘The’ to the start of this sentence.

Line 127. Please change to ‘inefficient’.

Line 131. Please change ‘prevents’ to ‘prevent’.

Line 136 - please change to ‘the DNA methylation landscape’.

Line 139 - formatting error of reference 23.

Line 143. Change ‘exists’ to ‘exist’.

Lines 219 and 463. Arabidopsis should have a capital letter.

Reviewer #3: The manuscript of Graindorge et al. addresses the question how UV induced DNA damage influences the epigenomic landscape of Arabidopsis. For me this is a novel and very innovative question of general interest. The authors apply a row of genome wide approaches to clarify the situation. Indeed, they can show that different repair pathways are required for keeping the epigenome stable and that document asymmetric epigenetic changes due to repair. They could detect chromocenter reorganization due to UV-C and methylation changes in constitutive and facultative heterochromatin. They analyzed methylation pattern changes in three different DNA repair pathways, (direct, NER and sRNA-mediated). They also analyze the genome wide production of photoproducts in WT and mutant backgounds. Here they find more damage in centromeric and pericentromeric regions. Moreover, they can show that in 2% of the damaged sites un-proper DNA methylation patters are induced due to repair. Based in their mutant analysis they suggest that DDB2 acts as general regulator of de novo methylation during development and stress. They speculate that the UV induced modulation of the DDB2 pool might lead to misregulation of de novo methylation.

The paper is not an easy read, as the authors present huge amounts of data, which gives the manuscript a quite descriptive flavor. I would suggest the authors to shorten the text, especially in the results and but also the discussion section concentrating on their most important findings and conclusions. This would make the whole story much more digestible also to the general reader.

**Have all data underlying the figures and results presented in the manuscript been provided?**

Reviewer #1: Yes

Reviewer #2: Yes

Reviewer #3: Yes

PLOS authors have the option to publish the peer review history of their article (what does this mean?). If published, this will include your full peer review and any attached files.

Reviewer #1: No

Reviewer #2: No

Reviewer #3: No

---

## [Decision Letter · Decision Letter 1]

3 Oct 2019

Dear Dr Molinier,

Thank you very much for submitting your revised Research Article entitled 'Photodamage repair pathways contribute to the accurate maintenance of the DNA methylome landscape upon UV exposure' to PLOS Genetics. It was reviewed by two of the original three reviewers. While one of these reviewers now recommends publication, the other does not consider that you satisfactorily addressed their original comments. We would like to give you one last opportunity to respond to the remaining criticisms of Reviewer 2 before we make a final editorial decision.

Should you decide to revise the manuscript for further consideration here, please address each of the specific points made by the reviewer. We will also require a detailed list of your responses to the review comments and a description of the changes you have made in the manuscript.

If you decide to revise the manuscript for further consideration at PLOS Genetics, please aim to resubmit within the next 60 days, unless it will take extra time to address the concerns of the reviewers, in which case we would appreciate an expected resubmission date by email to plosgenetics@plos.org.

[LINK]

We are sorry that we cannot be more positive about your manuscript at this stage. Please do not hesitate to contact us if you have any concerns or questions.

Yours sincerely,

Julian E. Sale

Associate Editor

PLOS Genetics

Wolf Reik

Section Editor: Epigenetics

PLOS Genetics

Reviewer's Responses to Questions

**Comments to the Authors:**

Reviewer #2: I think the reviewers for their attention to my comments. However, most of the questions that cause me most concern remain incompletely or unsatisfactorily answered.

First the question of correlation between your replicates. Please answer my question directly. For the replicates what is the correlation value between them? In the response the authors state that ‘two biological replicates were pooled… therefore correlation test cannot be applied’. Im afraid that doesn’t make sense - two samples can easily be correlated. Please answer the question.

The authors are resistant to plotting methylation along the chromosomes as requested, instead focusing on DMRs. DMRs are fine, but I don’t see why methylation could not be plotted in the requested way. If it doesn’t look any different then that provides a clear indication of the magnitude of the effects at a genome wide scale.

For one of my points I requested that the authors provide screenshots of their data shown in a genome browser. They provide for example in Figure S4 a plot around the ONSEN transposon, which is supposed to represent the trends reported genome wide. On this plot is highlighted DMRs by red rectangles. I want to see the data that the DMR is based on - I would suggest plotting the methyl value for each cytosine across this region as a histogram, i.e. the data that the DMR locations is based on. Equally, the siRNA data shown do not seem to be changing between the genotypes shown or between the time points in any of the single genotypes. I'm afraid if this is the ‘best’ locus the authors can provide it doesn’t fill me with a great deal of confidence about the data that underlies the conclusions being drawn.

Overall, I'm sorry to say that I find the conclusions are not supported by the data and analysis presented, and as such I cannot support publication.

Reviewer #3: The athours imporved the manusript, which in my opinion is now accepable for publication.

**Have all data underlying the figures and results presented in the manuscript been provided?**

Reviewer #2: Yes

Reviewer #3: None

PLOS authors have the option to publish the peer review history of their article (what does this mean?). If published, this will include your full peer review and any attached files.

Reviewer #2: No

Reviewer #3: No

---

## [Editor Report · Decision Letter 2]

13 Oct 2019

Dear Dr Molinier,

We are pleased to inform you that your manuscript entitled "Photodamage repair pathways contribute to the accurate maintenance of the DNA methylome landscape upon UV exposure" has been editorially accepted for publication in PLOS Genetics. Congratulations!

Yours sincerely,

Julian E. Sale

Associate Editor

PLOS Genetics

Wolf Reik

Section Editor: Epigenetics

PLOS Genetics

Comments from the reviewers (if applicable):

**Data Deposition**

http://datadryad.org/submit?journalID=pgenetics&manu=PGENETICS-D-19-01074R2

**Press Queries**

---

## [Editor Report · Acceptance letter]

29 Oct 2019

PGENETICS-D-19-01074R2 

Photodamage repair pathways contribute to the accurate maintenance of the DNA methylome landscape upon UV exposure 

Dear Dr Molinier, 

We are pleased to inform you that your manuscript entitled "Photodamage repair pathways contribute to the accurate maintenance of the DNA methylome landscape upon UV exposure" has been formally accepted for publication in PLOS Genetics! Your manuscript is now with our production department and you will be notified of the publication date in due course.

With kind regards,

Nicholas White

PLOS Genetics

On behalf of:
